# Multistage and transmission-blocking targeted antimalarials discovered from the open-source MMV Pandemic Response Box

Janette Reader[1], Mariëtte E. van der Watt[1], Dale Taylor[2], Claire Le Manach [2], Nimisha Mittal[3], Sabine Ottilie[3], Anjo Theron [4], Phanankosi Moyo [1], Erica Erlank[5], Luisa Nardini [5], Nelius Venter[5], Sonja Lauterbach [6], Belinda Bezuidenhout[6], Andre Horatscheck [2], Ashleigh van Heerden [1], Natalie J. Spillman [7], Anne N. Cowell[3], Jessica Connacher[1], Daniel Opperman[1], Lindsey M. Orchard [8], Manuel Llinás [8,9], Eva S. Istvan [7], Daniel E. Goldberg [7], Grant A. Boyle[2], David Calvo [10], Dalu Mancama[4], Theresa L. Coetzer [6], Elizabeth A. Winzeler [3], James Duffy[11], Lizette L. Koekemoer [5], Gregory Basarab [2], Kelly Chibale[2,12] & Lyn-Marié Birkholtz [1✉]

Chemical matter is needed to target the divergent biology associated with the different life cycle stages of *Plasmodium*. Here, we report the parallel de novo screening of the Medicines for Malaria Venture (MMV) Pandemic Response Box against *Plasmodium* asexual and liver stage parasites, stage IV/V gametocytes, gametes, oocysts and as endectocides. Unique chemotypes were identified with both multistage activity or stage-specific activity, including structurally diverse gametocyte-targeted compounds with potent transmission-blocking activity, such as the JmjC inhibitor ML324 and the antitubercular clinical candidate SQ109. Mechanistic investigations prove that ML324 prevents histone demethylation, resulting in aberrant gene expression and death in gametocytes. Moreover, the selection of parasites resistant to SQ109 implicates the druggable V-type $H^+$-ATPase for the reduced sensitivity. Our data therefore provides an expansive dataset of compounds that could be redirected for antimalarial development and also point towards proteins that can be targeted in multiple parasite life cycle stages.

[1] Department of Biochemistry, Genetics and Microbiology, Institute for Sustainable Malaria Control, University of Pretoria, Hatfield, Pretoria 0028, South Africa. [2] Drug Discovery and Development Centre (H3D), University of Cape Town, Rondebosch, Cape Town 7701, South Africa. [3] Division of Host-Microbe Systems & Therapeutics, Department of Pediatrics, University of California San Diego, La Jolla, CA 92093-076, USA. [4] Next Generation Health, Council for Scientific and Industrial Research, Pretoria 0001, South Africa. [5] Wits Research Institute for Malaria, School of Pathology, Faculty of Health Sciences, University of the Witwatersrand, and Centre for Emerging Zoonotic and Parasitic Diseases, National Institute for Communicable Diseases of the National Health Laboratory Service, Johannesburg 2193, South Africa. [6] Department of Molecular Medicine and Hematology, Wits Research Institute for Malaria, School of Pathology, Faculty of Health Sciences, University of the Witwatersrand, Johannesburg 2193, South Africa. [7] Division of Infectious Diseases, Department of Medicine, Washington University, St. Louis, MO 63110, USA. [8] Department of Biochemistry & Molecular Biology and the Huck Centre for Malaria Research, Pennsylvania State University, University Park, PA 16802, USA. [9] Department of Chemistry, Pennsylvania State University, University Park, PA 16802, USA. [10] Global Health Incubator Unit, GlaxoSmithKline (GSK), Severo Ochoa, 2, 28760 Tres Cantos, Madrid, Spain. [11] Medicines for Malaria Venture, International Center Cointrin, Route de Pré-Bois 20, 1215 Geneva, Switzerland. [12] South African Medical Research Council, Drug Discovery and Development Research Unit, Department of Chemistry and Institute of Infectious Disease and Molecular Medicine, University of Cape Town, Rondebosch 7701, South Africa. ✉email: lbirkholtz@up.ac.za

Malaria treatment solely relies on drugs that target the parasite, but current treatment options have a finite lifespan due to resistance development. Moreover, while current antimalarials are curative of asexual blood stage parasitaemia and associated malaria symptoms, they cannot all be used prophylactically and typically do not effectively block transmission. The inability to block transmission is limiting malaria elimination strategies, which requires chemotypes that block human-to-mosquito (gametocyte and gametes) and mosquito-to-human (sporozoites and liver schizonts) transmission.

As few as 100 sporozoites are able to initiate an infection after migrating to the liver for exoerythrocytic schizogony. The subsequent release of thousands of daughter cells, which in turn infect erythrocytes, initiates the extensive population expansion that occurs during asexual replication. A minor proportion (~1%)[1] of the proliferating asexual parasites will undergo sexual differentiation to form mature stage V gametocytes, a 10–14-day process in the most virulent parasite Plasmodium falciparum. Only ~$10^3$ of these falciform-shaped mature gametocytes are taken up by the next feeding mosquito to transform into male and female gametes in the mosquito's midgut[2]. Fertilization results in zygote development, and a motile ookinete that passes through the midgut wall forms an oocyst from which sporozoites develop, making the mosquito infectious.

The sporozoite and gametocyte population bottlenecks[3] have been the basis of enticing arguments for developing chemotypes able to target them. However, most compounds able to kill asexual parasites are either ineffective in preventing infection and/or blocking transmission or are compromised by resistance development (e.g. antifolates active as prophylactics) or toxicity concerns (e.g. primaquine targeting gametocytes with associated haemolytic toxicity in glucose-6-phosphate dehydrogenase-deficient patients). Patients treated with current antimalarials (or asymptomatic carriers) may harbour enough gametocytes to be transmitted to mosquitoes and sustain the malaria burden. The development of gametocyte-targeted transmission-blocking compounds is therefore essential for a complete strategy directed at eliminating malaria.

Phenotypic screens of millions of compounds have successfully identified antimalarial hits to populate the drug discovery pipeline. However, the majority of these screens assessed activity against asexual blood stage parasites as the primary filter, and hits were only profiled thereafter for activity against additional life cycle stages. While this strategy can identify compounds targeting two or more life cycle stages, it does not allow de novo discovery of compounds with selective activity against specific life cycle stages, such as gametocytes. Parallel screening against multiple life cycle stages would best identify such compounds and rely on selective and predictive assays for gametocytocidal activity[4], transmission-blocking[5,6], and hepatic development[7]. Recently, parallel screening of diversity sets has resulted in reports of such stage-specific compounds[7–10], with the benefit that divergent biology associated with the different life cycle states can be targeted[5,6,11].

In this work, we describe the parallel screen of the Medicines for Malaria Venture (MMV) Pandemic Response Box (PRB) on Plasmodium asexual stage parasites, liver stage parasites, mature (stage IV/V) gametocytes, male gametes and oocysts (Fig. 1), as well as mosquito endectocide activity. The MMV, in partnership with the Drugs for Neglected Disease Initiative, assembled the PRB as a collection of 400 drug-like compounds stratified by antibacterial, antiviral or antifungal activity (201, 153, and 46 compounds, respectively), with some compounds having antineoplastic activity. The unique and diverse nature of the compounds in the PRB allow one to explore and target the unique biology in the parasite's different life cycle stages to identify chemical starting points for antimalarial development. All screens were performed on the human parasite P. falciparum, except for the liver stages, where the established Plasmodium berghei assay was used[12,13]. Hit selection and progression of compounds in our screening cascade was not biased towards activity on any single life cycle stage, allowing the discovery of multistage-active scaffolds and those with stage-specific activity. Importantly, we report the profiling of a subset of compounds as transmission-blocking molecules that would not have been identified in a test cascade that began solely with an asexual blood stage assay. Four transmission-targeted leads include compounds that are chemically tractable, have favourable physicochemical properties and operate by previously undescribed modes of action, making them amenable to development as transmission-blocking antimalarials.

## Results

**Parallel screening of the PRB reveals hits against multiple life cycle stages.** To identify active compounds against different stages of the Plasmodium life cycle (irrespective of their activity against the other life cycle stages), the PRB was screened in parallel against PfNF54 asexual parasites and stage IV/V gametocytes and P. berghei liver stage (Fig. 1). The PfNF54 stage IV/V gametocyte data were orthogonally validated on three independent gametocyte assays to confirm that hit selection (compounds active on at least two platforms) was independent of assay readout[4]. Hits were identified with a relatively lenient but inclusive cut-off of ≥50% inhibition at 2 μM for asexual stages (within 4 S.D.) and 5 μM for gametocytes and liver stages (both within 3 S.D.). Asexual stage activity was confirmed against drug-resistant PfDd2 asexual parasites. Cytotoxicity filtering was applied after evaluation of the 50% inhibitory concentration ($IC_{50}$), and transmission-blocking potential of compounds with gametocytocidal activity was confirmed by inhibition of male gamete exflagellation and in a standard membrane feeding assay (SMFA). Assay reproducibility indicators are provided in Supplementary Table 1.

An 18% hit rate was obtained against PfNF54 asexual parasites, 12% against PfNF54 stage IV/V gametocytes and 11% against liver stage parasites (Fig. 2a, Supplementary Data 1). Although a number of compounds showed activity against all these life cycle stages, stage-specific differentiation was evident, as exemplified by the overrepresentation of antifungal compounds in the hit pool for stage IV/V gametocytes compared to asexual parasites (Fig. 2b). The remaining hits reflect the distribution of the compounds in the PRB, with the highest number of hits classified as antibacterials followed by antivirals. The latter seemed to be more potent (as a percentage of the hits) on PfNF54 stage IV/V gametocytes relative to PfNF54 asexual parasites. Only four compounds showed marked toxicity against Chinese hamster ovarian (CHO) cells (<50% viability at 2 μM, supplementary Fig. 1). Superimposition of the PRB chemical space on current antimalarial drugs within the launched drugs chemical space (Supplementary Fig. 2) indicates little overlap.

Based on the target indicators/biological pathway descriptors for the PRB compounds in other diseases, PfNF54 asexual hits were enriched for inhibitors of kinases, CYP450, energy metabolism and DNA synthesis. Inhibitors of dihydrofolate reductase (DHFR, antifolates), dihydroorotate dehydrogenase, proton pumps and topoisomerase were exclusive hits for PfNF54 asexual and liver stage parasites. Compounds with both PfNF54 asexual and gametocyte activity include antithrombotics, protease inhibitors, sphingosine-1-phosphate receptor modulators and compounds affecting redox homoeostasis, whereas inhibitors of the mycobacterial MmpL3 (mycobacterial membrane protein large 3) and ion channels were predominant in gametocyte hits (Fig. 2c). Chemical classes highly represented in the hit pool include quinolines, benzamides/benzoids and azoles.

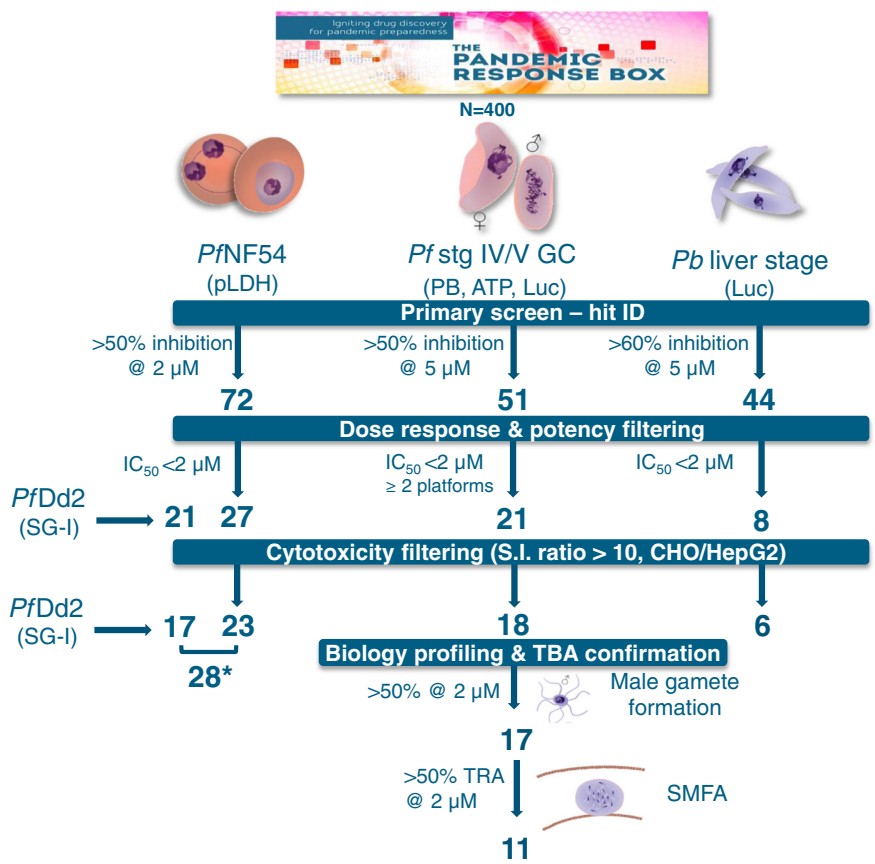

**Fig. 1 Screening cascade of the MMV Pandemic Response Box for activity against multiple life cycle stages of *Plasmodium*.** The 400 compounds in the PRB were screened in a primary assay against drug-sensitive (NF54) *P. falciparum* asexual blood stages (ABS, at 2 and 20 μM) and mature gametocytes (stage IV/V, GC, 1 and 5 μM) and *P. berghei* liver stages (5 μM). Hits were selected based on ≥50% inhibition at specific concentrations as indicated. The criteria for each decision point are indicated followed by the number of compounds that passed the criteria. Compounds were additionally evaluated in dose response on drug-resistant asexual Dd2 parasites (chloroquine, pyrimethamine and mefloquine resistant). * = the potent ABS compounds (IC$_{50}$ < 2 μM), after removing toxic compounds and eliminating overlapping compounds between *Pf*NF54 and *Pf*Dd2, amount to 28 compounds in total. IC$_{50}$ 50% inhibitory concentration, pLDH parasite lactate dehydrogenase assay, PB PrestoBlue® assay, ATP ATP viability assay, Luc luciferase reporter lines assays, *Pf Plasmodium falciparum*, *Pb Plasmodium berghei*, S.I. selectivity index, CHO Chinese hamster ovarian cells, HepG2 hepatocellular carcinoma line, TBA transmission-blocking activity, TRA transmission-reducing activity, SMFA standard membrane feeding assay. Parasite drawings were modified from freely available images (https://smart.servier.com/), under a Creative Commons Attribution 3.0 Unported Licence.

**Multistage-active compounds**. All hit compounds were counter-screened against either CHO or HepG2 mammalian cells to remove cytotoxic compounds (Supplementary Data 2) from further consideration for characterization as antimalarials. Of the 72 PRB hits against asexual parasites, all activities were confirmed on re-screening, and of these, 28 compounds passed a further stringent cut-off with IC$_{50}$ values <2 μM (Fig. 2a and Supplementary Data 2), with 16 compounds exclusively active against the asexual stages (Fig. 3a). Of the 51 hits against *Pf*NF54 stage IV/V gametocytes, 18 compounds had IC$_{50}$ values <2 μM. Eight shared activity against asexual stages but ten had gametocyte stage-specific activity (Fig. 3a). Only six compounds showed activity against *P. berghei* liver stages (<2 μM). Notably, two compounds showed pan-reactivity against all life cycle stages: the peptidomimetic antitumour agent MMV1557856 (Birinapant, a SMAC-mimetic inhibitor of apoptosis proteins (IAPs))[14] and the imidazoquinoline antitumour agent MMV1580483 (AZD-0156, a Ataxia Telangiectasia Mutated kinase inhibitor) (Fig. 3b and Supplementary Table 2).

**Asexual-specific chemotypes**. Encouragingly, the 28 compounds with asexual parasite activity (Supplementary Data 2) included the known antimalarial compounds chloroquine (MMV000008) and

tafenoquine (MMV000043), which were both present in the PRB and showed IC$_{50}$s comparable to those previously reported[6] (30 and 940 nM, respectively), validating the screening process (Fig. 4). The most potent of the 28 compounds was an antibacterial diaminopyridine propargyl-linked antifolate[15], MMV1580844 (IC$_{50}$ = 0.0017 μM), which targets DHFR in mammalian and yeast cells[16,17], and was active against *P. berghei* liver stages (0.004 μM) but not *Pf*NF54 stage IV/V gametocytes, supporting previous reports that inhibition of *Plasmodium* DHFR is effective in asexual and liver stages[6]. MMV1580844 had a pronounced (63-fold) loss of activity against pyrimethamine-resistant *Pf*Dd2. By contrast, the quinazoline antifolate trimetrexate (MMV1580173, derived from methotrexate) was potently active only against *Pf*Dd2 (IC$_{50}$ = 0.108 μM) as well as *P. berghei* liver stages (IC$_{50}$ = 0.0005 μM), in both instances with >10-fold selectivity towards the parasite vs CHO cells.

Importantly, the majority (13/16) of the asexual-specific compounds had not been previously reported with antiplasmodial activity nor do they show structural similarity to any antiplasmodial (Fig. 4 and Supplementary Fig. 2), providing chemotypes for further evaluation. Interestingly, the majority of the asexual-specific compounds are classified as antibacterials and

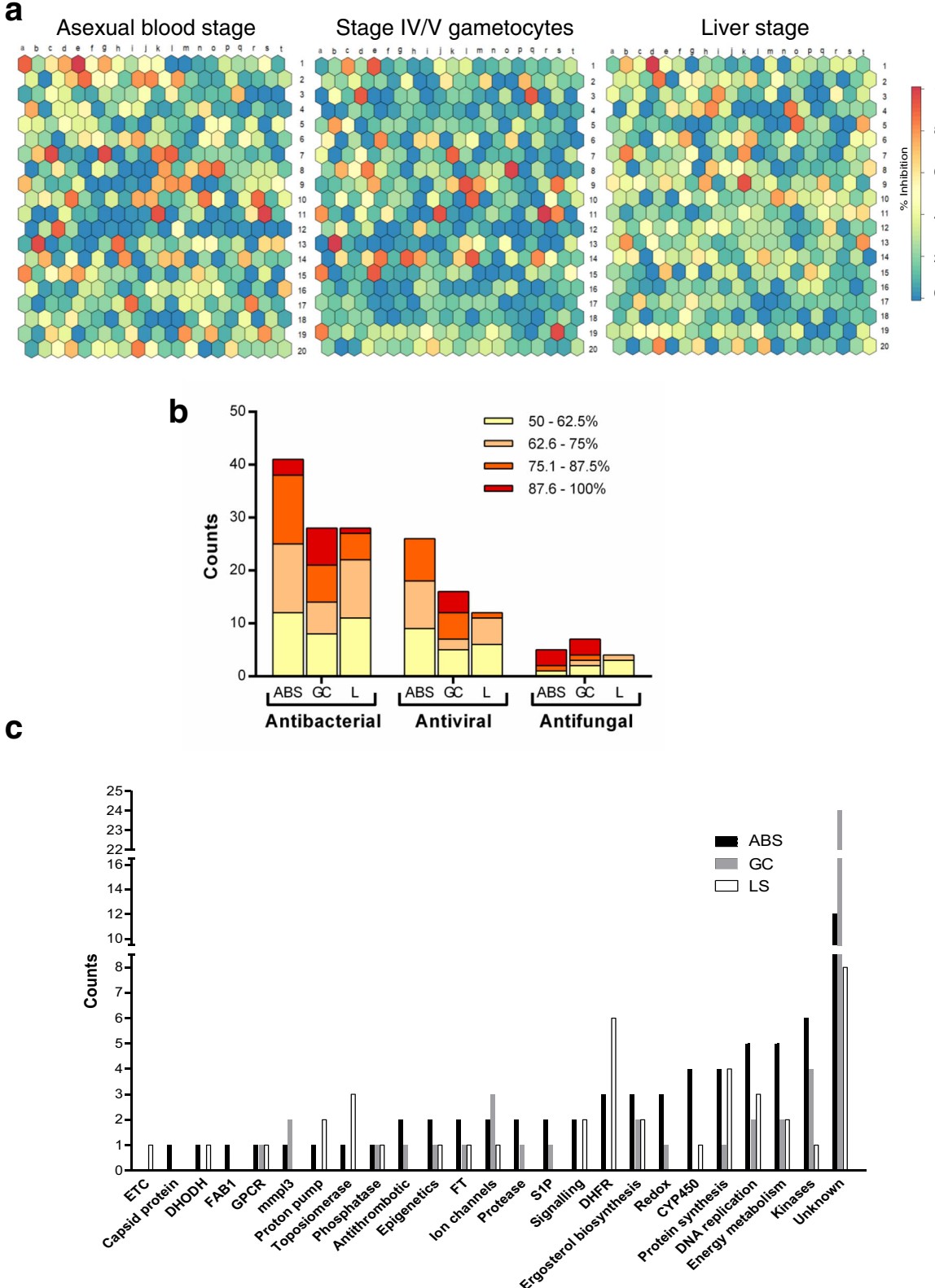

include two kinase inhibitors, MMV1593539 (IC$_{50}$ = 0.686 μM), a pyruvate kinase inhibitor, and MMV1580482 (URMC-099-C, IC$_{50}$ = 1.3 μM), a mixed lineage kinase 3 (MLK3) inhibitor. Additionally, there is pibenzimol (MMV020752, IC$_{50}$ = 0.149 μM), a disrupter of DNA replication, and MMV019724 (IC$_{50}$ = 1.67 μM), an antiviral lactate dehydrogenase inhibitor.

**Liver stage activity is associated with asexual parasite activity.** Six dual-stage compounds (asexual blood stage and *P. berghei* liver stage activity ≤2 μM) were identified in the PRB, marking them as having prophylactic potential (IC$_{50}$ range from 0.0005 to 1.72 μM, Fig. 4). These compounds include not only previously described antifolates (MMV1580173 and MMV1580844) but

**Fig. 2 Primary screening of the PRB for hits against *P. falciparum* parasites. a** Supra-hexagonal maps of all 400 compounds included in the PRB after analysis on *P. falciparum* NF54 asexual blood stage parasites, stage IV/V gametocytes, and *Pb* liver stage. Each hexagon is indicative of a single compound and the order of the hexagon is the same between the plots. Colours on the heat bar indicate percentage of inhibition of proliferation (asexual blood stage parasites) or viability (stage IV/V gametocytes) after treatment with each compound at either 2 µM (asexual blood stages) or 5 µM (stage IV/V gametocytes or liver stages), screened at least in duplicate. The data for the stage IV/V gametocytes are compiled from hits identified with three different assay platforms, run in parallel (ATP, PrestoBlue® and luciferase reporter expression) with any hit on any platform included, and where identified on >2 platforms, the highest value was included. **b** Proportional distribution of hits (>50% inhibition @ 2 µM for asexual blood stages or 5 µM for stage IV/V gametocytes and liver stages) based on disease area as defined in the PRB. Bars are delineated to show activity distribution. ABS asexual blood stages, GC stage IV/V gametocytes, L liver stage. **c** Stratification of hits from **a** (>50% inhibition @ 2 µM for asexual blood stages or 5 µM for stage IV/V gametocytes and liver stages, screened in duplicate). The number of compounds that could be associated with a specific biological activity or target indicator is shown. Protein targets/metabolic pathways were identified based on the descriptions of compounds with known activity in the PRB in other disease systems. FT farnesyltransferase inhibitors, GPCR G protein-coupled receptors, S1P sphingosine-1-phosphate receptor modulators, CYP cytochrome inhibitors, ETC electron transport chain, DHFR dihydrofolate reductase, DHODH dihydroorotate dehydrogenase.

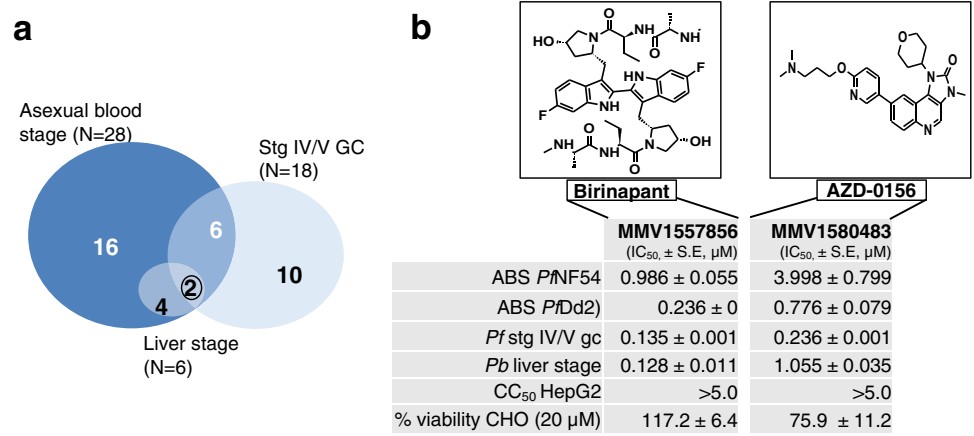

| | Birinapant | AZD-0156 |
|---|---|---|
| | **MMV1557856** (IC$_{50}$, ± S.E, µM) | **MMV1580483** (IC$_{50}$, ± S.E, µM) |
| ABS *Pf*NF54 | 0.986 ± 0.055 | 3.998 ± 0.799 |
| ABS *Pf*Dd2) | 0.236 ± 0 | 0.776 ± 0.079 |
| *Pf* stg IV/V gc | 0.135 ± 0.001 | 0.236 ± 0.001 |
| *Pb* liver stage | 0.128 ± 0.011 | 1.055 ± 0.035 |
| CC$_{50}$ HepG2 | >5.0 | >5.0 |
| % viability CHO (20 µM) | 117.2 ± 6.4 | 75.9 ± 11.2 |

**Fig. 3 Active compounds on multiple stages of *Plasmodium* development after dose–response evaluation and cytotoxicity filtering. a** Venn diagram of the number of compounds identified with activity (inhibitory concentration, IC$_{50}$) below 2 µM, for which no cytotoxicity was identified on either CHO cells (>50% viability at 2 µM or selectivity index >10) or HepG2 cells (selectivity index >10). **b** A total of two compounds with activity against all life cycle stages tested: Birinapant and AZD-0159. Asexual blood stage activity (ABS) was determined against both drug-sensitive (NF54) and drug-resistant (Dd2) *P. falciparum*. GC *P. falciparum* stage IV/V gametocytes. Toxicity indicated both at CC$_{50}$ (cytotoxic concentration) against HepG2 cells as well as for viability of CHO cells remaining after 20 µM treatment. Data are from three independent biological repeats, performed with minimum technical duplicates, mean ± S.E. 95% confidence intervals on all IC$_{50}$ values are provided in Supplementary Data 2.

also the ribonucleotide reductase inhibitor MMV1580496 (triapine) and the bacterial methionyl-tRNA synthetase inhibitor MMV1578884 (REP3123) in addition to the multistage-active compounds, AZD-0156 and Birinapant.

**Unique compounds with stage-specific activity against IV/V gametocytes.** Eighteen compounds were active (IC$_{50}$s < 2 µM) against late-stage gametocytes, as confirmed in at least two of the three orthogonal assays (ATP, PrestoBlue® or luciferase reporter assays, Supplementary Data 2). Of these, eight compounds shared asexual parasite activity, but importantly, ten compounds selectively inhibited *Pf*NF54 stage IV/V gametocytes (>10-fold difference in IC$_{50}$, Fig. 5). The most potent compound was the antineoplastic epidrug ML324[18] (MMV1580488, IC$_{50}$ = 0.077 µM), and a marked selection for structurally unrelated inhibitors of G protein-coupled receptors (GPCRs) and related transmembrane proteins were present (Fig. 5a). This includes MMV1581558 (IC$_{50}$ = 0.130 µM) and two MmpL3 inhibitors: the well-characterized 1,2-ethylene diamine antitubercular clinical candidate SQ109[19–21] (MMV687273, IC$_{50}$ = 0.105 µM) and a rimonabant derivative MMV1580843[22] (IC$_{50}$ = 0.108 µM). Two structurally distinct imidazole antifungals also showed potent activity against gametocytes (Supplementary Fig. 3):

MMV1634491 (IC$_{50}$ = 0.208 µM) and MMV1634492 (the topical antifungal eberconazole, IC$_{50}$ = 0.23 µM).

**Gametocytocidal compounds target male gametes.** Gametocytocidal hits were validated on male gametes that are generally more sensitive to compounds[23]. The majority of compounds (13) inhibited male gamete exflagellation by >60% (2 µM, Fig. 5b), 8 of which were potent at ≥80% inhibition. The latter included the gametocyte-targeted compounds MMV1580488 (ML324), the azole antifungals MMV1634491 and MMV1634492 and the MmpL3 inhibitor MMV687273 (SQ109), as well as compounds with additional asexual parasite activity (MMV1580483, MMV396785, MMV1582495, MMV1578570, MMV1581558). The gametocytocidal activity was irreversible for all the compounds except MMV1578570 (Supplementary Fig. 4)[24]. Six compounds (the epidrug MMV1580488, the two azole antifungals (MMV1634491, MMV1634492), a quinoline MMV1634399 and two MmpL3 inhibitors (MMV1580843, MMV687273)) were directly active on male gametes, implying that shared essential biology between these stages is being targeted.

**Chemotypes with confirmed transmission-blocking activity (TBA).** Transmission-blocking activity was validated by

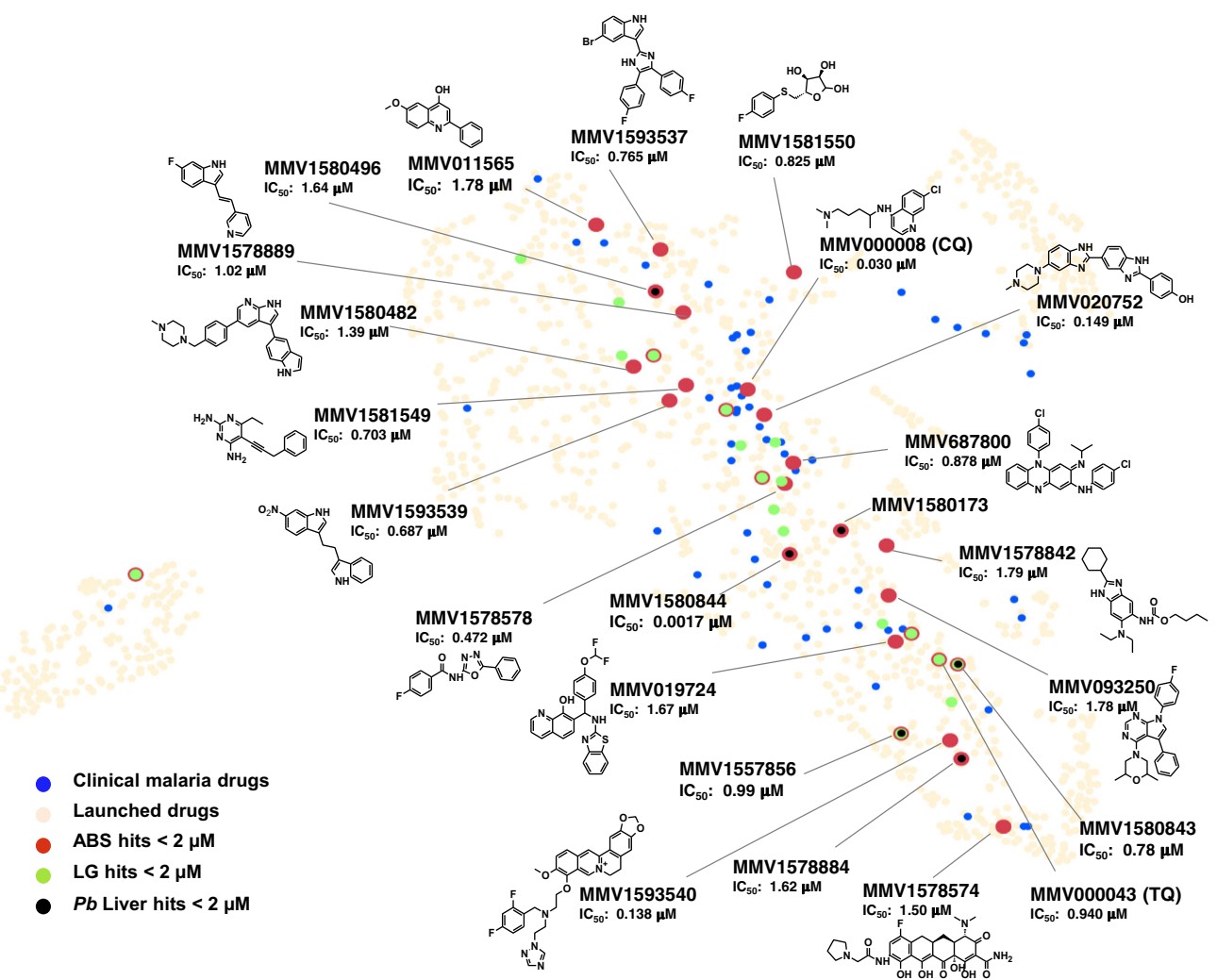

**Fig. 4 Asexual blood stage active compounds from the PRB in relation to malaria clinical drugs.** PRB compounds active against asexual blood stage (ABS) parasites of *P. falciparum*. Compounds with inhibitory concentrations (IC$_{50}$) <2 μM were identified as hits against either *Pf*NF54 or *Pf*Dd2. PRB hits are represented in the launched drugs chemical space (available within the StarDrop V 6.6 software, beige) in comparison to malaria clinical drugs (blue dots). Asexual blood stage actives with IC$_{50}$ values <2 μM are indicated in red, and gametocyte actives and liver stage actives (all at the same cut-off) are indicated in green and black, respectively, with different dot diameters to highlight compounds active on multiple stages. The 16 asexual-specific compounds are labelled with Compound ID, asexual stage IC$_{50}$ and structure and other compounds of interest just by name and IC$_{50}$. CQ chloroquine, TQ tafenoquine.

the mosquito-based SMFA[25–27], using an African malaria vector, *Anopheles coluzzii* (G3). Transmission-reducing activity (TRA; reduction in oocyst intensity) and TBA (reduction in oocyst prevalence) were determined 8–10 days after feeding female mosquitoes on gametocyte-infected blood, treated with selected compounds (2 μM, Fig. 5c, Supplementary Fig. 5 and Supplementary data 3). Total sample size for the control feeds averaged at 53 mosquitoes, with average oocyst prevalence at 71% and oocyst intensity of 5.8 oocysts/midgut. The TRA for MMV000043 (Tafenoquine) at 77% correlates with previous reports on *Anopheles stephensi*[28], validating assay performance with *A. coluzzii*. All the compounds (except for MMV1580853) reduced TRA by >60% and, remarkably, nine compounds had ≥80% TRA. Four gametocyte-targeted, structurally dissimilar compounds MMV1006203 (1,1-dioxide 1-Thioflavone), the azole antifungal MMV1634492, a quinoline MMV1634399 and the GPCR inhibitor MMV1581558 were able to block transmission (TBA) by ≥60% (MMV1634492 by 79%), associated with a significant reduction in oocyst intensity (*p* < 0.05, Supplementary Data 3).

**Endectocide activity**. Gametocytocidal compounds were evaluated for their activity as endectocides, killing mosquitoes after supplied in an uninfected blood meal. However, none of the compounds produced significant mortality (one-way analysis of variance (ANOVA), *p* = 0.7005, total DF = 71, *n* ≥ 2) in the 4-day mortality assay at 2 μM (Supplementary Fig. 6). Rather, moderate killing (~30%) was observed for the two MmpL3 inhibitors MMV687273 and MMV1580843, the GPCR inhibitor MMV1581558, AZD-0156 (MMV1580483) and MMV1174026, marking these compounds with some potential to kill the parasite and mosquito vector.

**Mechanistic explorations of transmission-targeted compounds MMV1580488 (ML324) and MMV687273 (SQ109)**. We investigated the potential mechanism of action of the most potent transmission-targeted compound, ML324, and the antitubercular clinical candidate SQ109.

ML324, together with another inhibitor JIB-04, was recently described as active site inhibitors of a recombinantly expressed

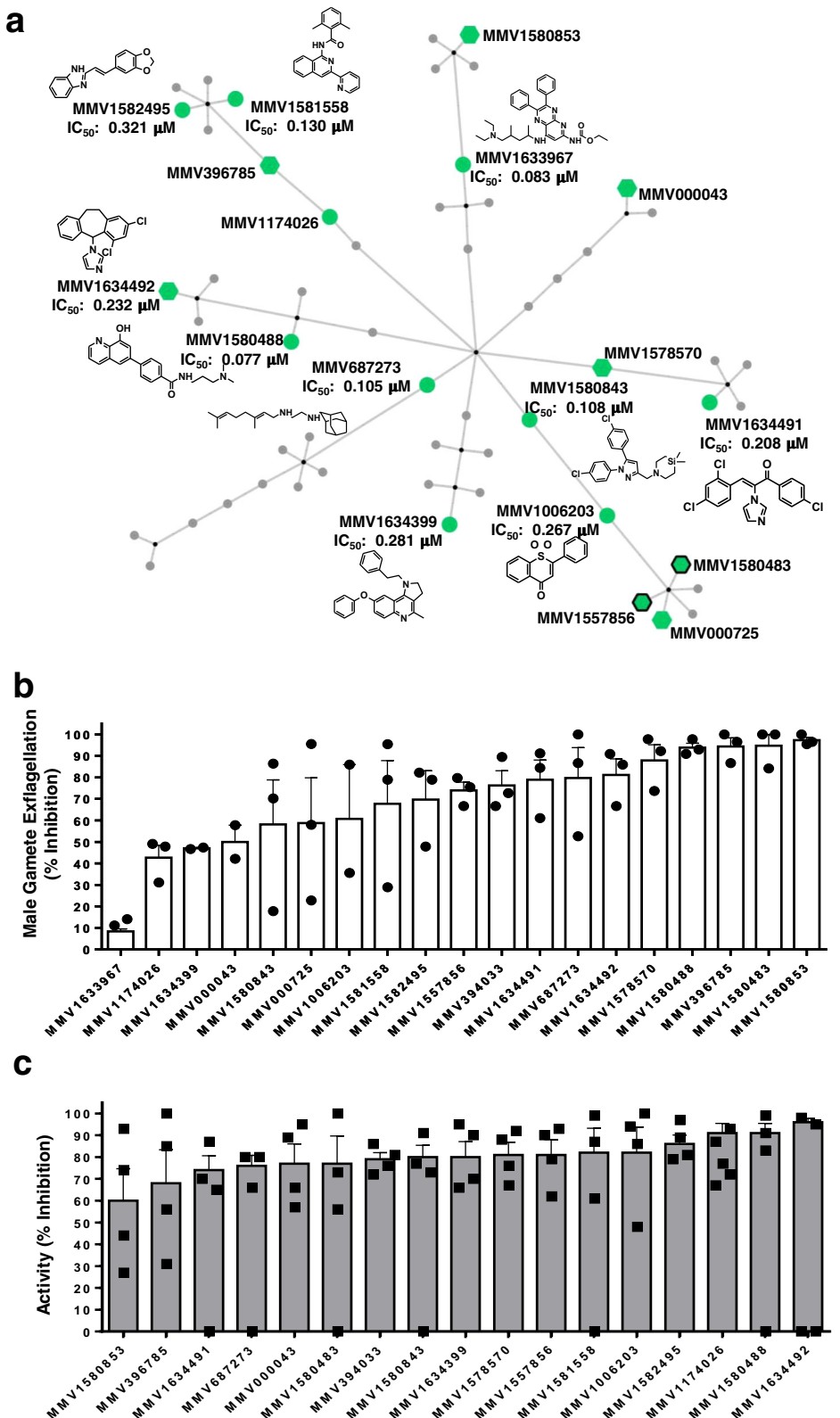

Jumonji-domain containing demethylase (KDM4), jmj3 in *Plasmodium*[29]. Inhibition of *Pf*jmj3 with JIB-04 results in altered histone methylation[29], aberrant gene expression and ultimately parasite death. Here we show that ML324 was significantly more potent against late-stage gametocytes compared to early-stage gametocytes and asexual parasites (Fig. 6a). Mechanistically,

ML324 treatment significantly increased H3K9me3 levels (a repressive heterochromatin mark in *P. falciparum* gameto-cytes)[30] in late-stage gametocytes, while not affecting acetylation (Fig. 6b, $p = 0.028$, $t$ test, $n = 3$). This translated to differential expression of 13% of the genome (Fig. 6c, $p < 0.01$, >0.5 $\log_2$ fold change (FC) in either direction, Supplementary Data 4), similar to

**Fig. 5 Transmission-blocking activity of active compounds from the PRB against *P. falciparum* stage IV/V gametocytes, gametes and oocysts.**
**a** Chemical cluster analysis of the gametocyte hit compounds, using the *FragFP* descriptor and a Tanimoto similarity index >0.50 in OSIRIS DataWarrior v 5.0.0, and network construction with Cytoscape v 3.7.2. Edges were assigned between similar scaffolds and a parent node. Active compounds with $IC_{50}$ values <2 µM are indicated in green, those with additional activity at the same cut-off on ABS are indicated with hexagons and those with shared activity on liver stages with black borders. Structures are highlighted for selected compounds. Data are from three independent biological repeats ($n = 3$), each performed in technical triplicates, ±S.E. **b** Nineteen compounds with activity against *P. falciparum* stage IV/V gametocytes were evaluated for their ability to inhibit male gamete exflagellation. Compounds (2 µM) were used on stage IV/V gametocytes for a 48-h treatment prior to inducing male gamete exflagellation (carry-over format). Data are from three independent biological repeats ($n = 3$), performed in technical triplicates, mean ± S.E indicated, except for MMV1633967, MMV1634399 and MMV000043, for which data are from $n = 2$ biological repeats). Individual data points are indicated in symbols. **c** SMFA data for 17 compounds (selected based on >50% inhibition on male gamete exflagellation). SMFA was performed by feeding *A. coluzzii* mosquitoes with compound-treated gametocyte cultures (48 h treatment at 2 µM). Data are presented as percentage of TRA (transmission-reducing activity, reduction in oocyst intensity: $\frac{Ci-Ti}{Ci} * 100$, where $i$: oocyst number (intensity), $C$: control and $T$: treated) from at least three independent biological repeats ($n = 4$), performed with technical duplicates, mean ± S.E indicated. Individual data points are indicated in symbols.

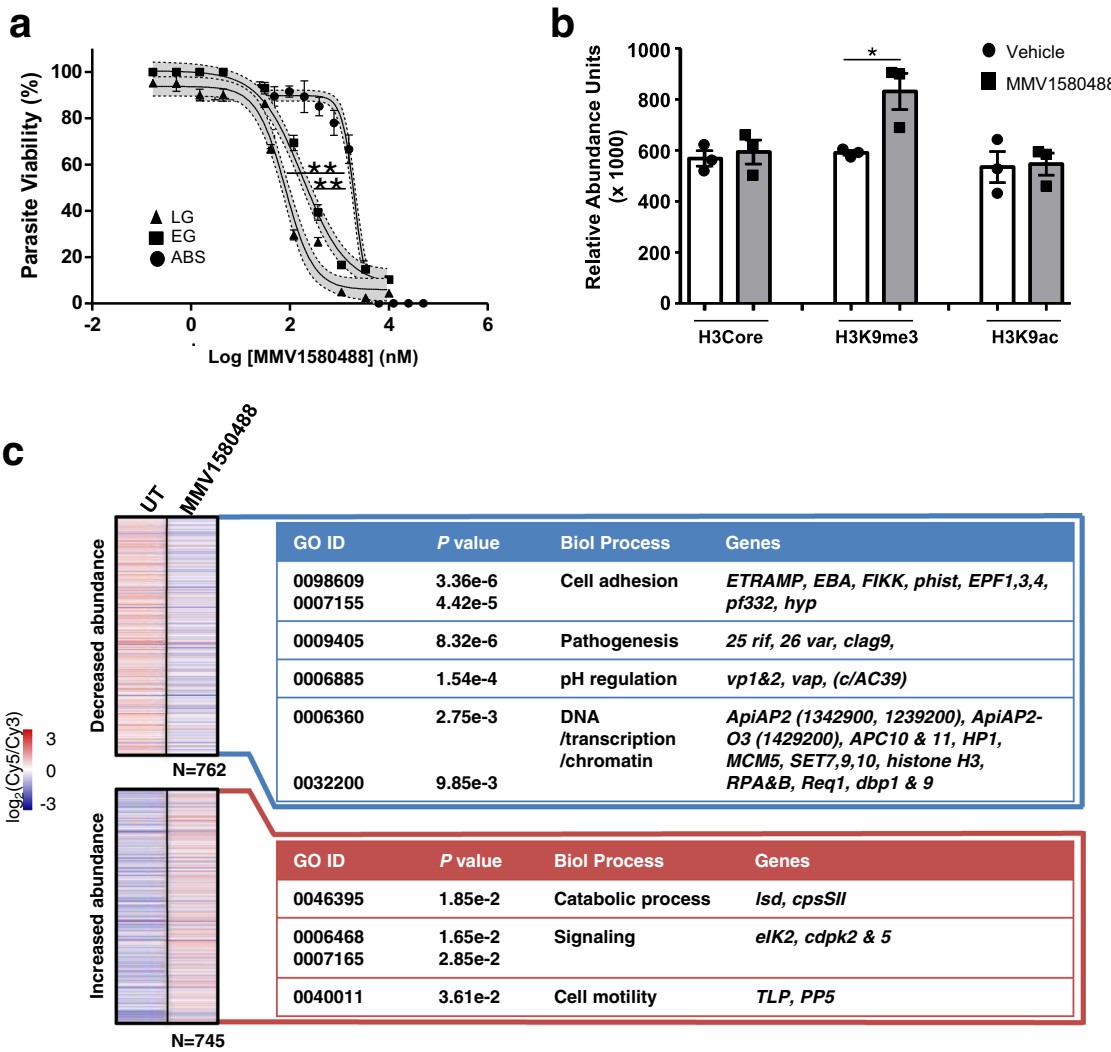

**Fig. 6 MMV1580488 (ML324) inhibition of histone demethylation and induction of aberrant gene expression in gametocytes. a** Dose–response analysis of ML324 on asexual parasites (ABS) and early- (stage II/III, EG) or late-stage (LG, stage IV/V) gametocytes on the PrestoBlue® assay. Data are from three independent biological repeats ($n = 3$), performed in technical triplicates, mean ± S.E indicated. 95% confidence intervals on each point indicated as ribbons for each sigmoidal curve. $IC_{50}$ ABS: 2.06 ± 0.119 µM; $IC_{50}$ EG: 0.188 ± 0.029 µM; $IC_{50}$ LG: 0.077 ± 0.001 µM; **$p = 0.0036$ ABS vs LG; **$p = 0.0052$ ABS vs EG, paired, two-tailed *t* test. **b** Relative abundance of histone modifications (anti-H3core, anti-H3K9me3 and anti-H3K9ac) for gametocytes treated with ML324 (5 µM) for 24 h. Data are from three independent biological repeats ($n = 3$), performed in technical triplicates, mean ± S. E indicated. *$p = 0.028$, paired, two-tailed *t* test. **c** Transcriptome analysis of gametocytes treated ML324 (5 µM) for 24 h indicating transcripts with significant differential expression ($\log_2$ FC > 0.5 increased (red) or decreased (blue) compared to control untreated parasites). GO annotations on biological process level indicated for the increased or decreased abundance gene sets (Fisher's exact test, two-tailed, exact *p* values provided on figure per GO term; gene names indicated in Supplementary Data 4).

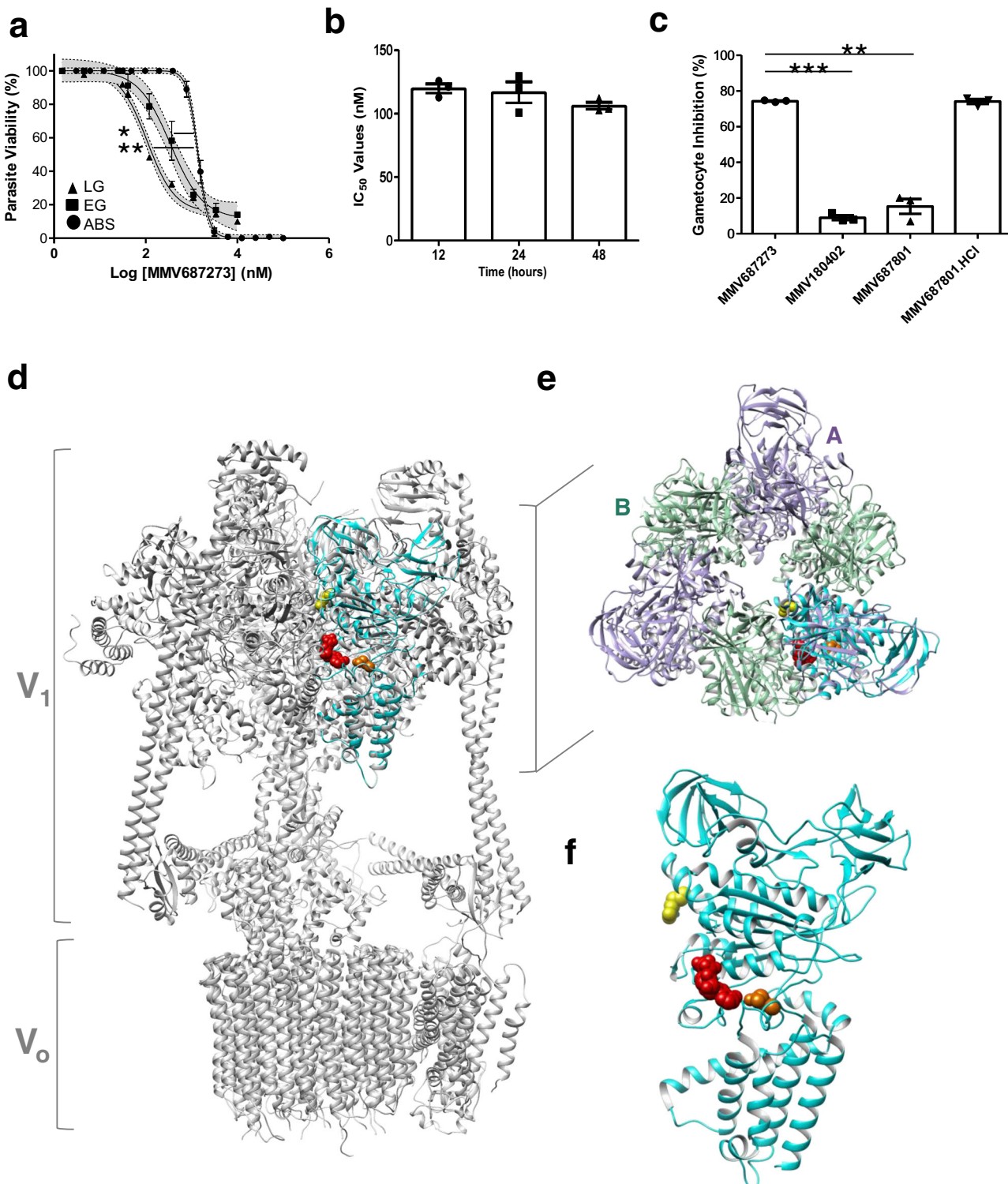

JIB-04 inhibition of *Pf*jmj3[29]. Despite differences in parasite stages evaluated for aberrant gene expression profiles due to JIB-04 or ML324 treatment (asexual vs gametocyte treatments), several processes were similarly affected by these two inhibitors of *Pf*jmj3 (Supplementary Data 4). Comparatively, there is a complete lack of overlap between aberrant gene expression due to ML324 treatment and that caused by inhibition of a histone methyltransferase with BIX-01294 in gametocytes[31], implying at least some conserved and specific responses associated with *Pf*jmj3 inhibition. ML324 treatment of gametocytes caused

repression of known H3K9me3-associated genes involved in cell adhesion (Fig. 6c)[30] and DNA/chromatin-related processes, including three histone methyltransferases (SET7, 9 and 10), histone H3 and heterochromatin protein 1. Several Api-AP2 transcription factor family members also showed aberrant expression under ML324 pressure including Api-AP2-O3 (ookinete associated). Among the transcripts with increased abundance is another jmj family member, jmjC2[30], and gametocyte-associated proteins gametocyte development 1 (gdv1) and male gamete gene 1 (*mdv1*). The lack of removal of heterochromatic

**Fig. 7 Mechanistic investigations of MMV687273 (SQ109). a** IC$_{50}$ for MMV687273 (SQ109) on late-stage gametocytes (LG, stage IV/V), early-stage gametocytes (EG, stage II/III) or asexual parasites (ABS) tested on the same assay platform (PrestoBlue®) for 48 h drug pressure. Data are from three independent biological repeats ($n = 3$), performed in technical triplicates, mean ± S.E indicated. 95% confidence intervals on each point indicated as ribbons on each sigmoidal curve. IC$_{50}$ ABS: 1.39 ± 0.090 μM; IC$_{50}$ EG: 0.383 ± 0.108 μM; IC$_{50}$ LG: 0.109 ± 0.002 μM; **$p = 0.0052$ ABS vs LG; *$p = 0.036$ ABS vs EG, paired, two-tailed $t$ test. **b** IC$_{50}$ for MMV687273 (SQ109) tested on stage IV/V gametocytes after 12, 24 or 48 h drug pressure (PrestoBlue® assay). Data are from three independent experiments ($n = 3$), each in triplicate, ±S.E. **c** The percentage of inhibition of stage IV/V viability under 1 μM pressure of MMV687273 (SQ109) compared to MMV180402 (2-adamantanamine) or the parent compound ethambutol (MMV687801, $S,S$-ethambutol and ethambutol·HCl). Data are from three independent biological repeats ($n = 3$), in technical duplicates each, mean ± S.E indicated. ***$p = 0.0006$, **$p = 0.0052$, paired, two-tailed $t$ test. **d** Structure of the *P. falciparum* V-type H$^+$-ATPase subunit A (*Pf*vapA, PF3D7_1311900, cyan in all cases, code Q76NM6 Swiss Model Repository), using the mammalian V-type ATPase from rat brain (6vq6.1) as template. The soluble V$_1$ and membrane-associated V$_o$ domains of the mammalian complex is indicated. **e** Top view of the V$_1$ domain, indicating the overlap of *Pf*vapA (cyan) with subunit A of the mammalian complex (purple) compared to subunit B (turquoise) and **f** structure of *Pf*vapA alone with side view. Bound ADP is indicated in red, the mutations in *Pf*vapA is as follows: Q225K in orange, R353K in yellow.

H3K9me3 in ML324-treated parasites therefore results in the inability of the parasite to prepare for gametogenesis (concurring with ML324's efficacy against male gametes, Fig. 5b), since gene silencing is maintained.

SQ109 showed a significant >10-fold selectivity towards late-stage gametocytes compared to asexual parasites ($p = 0.0052$, $n = 3$ paired $t$ test, Fig. 7a). Since SQ109 is an established inhibitor of MmpL3 in mycobacteria[19–21] with typical rapid action[21,32], we evaluated the rate of gametocytocidal action of SQ109 for 12, 24 and 48 h (Fig. 7b). No significant difference ($p = 0.937$ and $p = 0.558$, respectively; one-way ANOVA, $n = 3$) was observed in the IC$_{50}$ values, indicating that the effect was present already within 12 h of exposure to gametocytes. Moreover, the activity of SQ109 was similar to the parent compound $S,S$-ethambutol.2HCl[20] (MMV687801.2HCl), whereas a structurally related 2-adamantanamine (MMV180402) was inactive (Fig. 7c). We subsequently evaluated the potential for MmpL3-like targets in *P. falciparum*. The parasite's genome does not predict any direct homologues to MmpL3 and only two members of the resistance-nodulation-division (RND)-superfamily (to which MmpL3 belongs) are present. This includes a Niemann–Pick type C1-related H$^+$/lipid symporter (*Pf*NCR1)[33]. However, treatment of a *Pf*NCR1 knock-down line with SQ109 did not confer any marked loss of activity, indicating that SQ109 does not likely interact with this protein in *Plasmodium* spp. (Supplementary Fig. 7). In an attempt to identify alternative molecular targets, resistant mutant selection was performed on *P. falciparum* 3D7 (A10 clone) under SQ109 pressure. Stable but low-level resistance (2–3-fold increase in IC$_{50}$ to SQ109 compared to the parental line) was obtained from two independent selections. Whole-genome sequencing of two clones from each selection indicated mutations in the *P. falciparum* V-type H$^+$-ATPase (*Pf*vapA, PF3D7_1311900), at one position each: Q225K and R353K (Supplementary table 3). These mutations occur in the A subunit, part of the soluble, ATP-hydrolytic V$_1$ domain of the protein complex. Structural analysis of *Pf*vapA based on homology (61% sequence identity) to the mammalian V-type ATPase subunit A indicated that both mutations fall within the nucleotide-binding domain of the protein (Fig. 7d–f and Supplementary Fig. 8). Importantly, the Q225K mutation sits within the conserved nucleotide binding pocket of the protein and results in a steric clash with ADP (Fig. 7f), at the catalytic B–A subunit interface, distinguished from the non-catalytic A–B interface (Fig. 7e). *Pf*vapA has been shown to be present on the parasite's plasma membrane and digestive vacuole[34], and inhibition thereof typically results in parasite death due to disturbance of ATP-dependent H$^+$ efflux and pH regulation[35,36].

## Discussion

The ability to quickly respond to pandemics has become of paramount importance, and compound sets like the PRB provide an essential tool to support rapid screening of diverse druggable compounds for potential repurposing. Indeed, antimalarials have previously been investigated as antineoplastics[37,38] and several antibiotics and antifungals have previously demonstrated anti-malarial activities[39,40]. Here we screened the PRB across multiple *Plasmodium* stages and identified chemical matter with anti-malarial activity not previously described, providing a useful resource to the research community for drug repurposing as well as lead matter for drug optimization.

Multistage activity is a preferential attribute for the next generation of antimalarials[41], but such compounds are rarely found in diversity library screens, in large part due to targeted screening approaches rather than parallel screening in multiple assays. We identified two non-cytotoxic multistage-active compounds in the PRB (Birinapant and AZD-0156) that could point to biological parsimony of conserved targets in all these stages, essential to the survival of the parasite such as inhibiting proteins involved in cellular stress responses by either inducing apoptosis or preventing DNA damage recovery responses. Interestingly, Birinapant preferentially killed *Plasmodium*-infected hepatocytes by reducing host cellular IAP[42].

The dual-active asexual and liver stage compounds identified in the PRB have the potential for prophylactic and chemoprotective utility (target candidate profile 4 (TCP-4)), in addition to being chemotherapeutically relevant (target candidate profile 1 (TCP-1))[41]. Though compounds targeting the same parasite protein in both liver and asexual stages have the associated risk of target-based resistance, the smaller number of parasites in the liver stage reduces this risk. Interesting targets could be ascribed for some of these TCP-1 and -4 dual-active compound including MMV1578884 (REP3123) that targets *Clostridium difficile* methionyl-tRNA synthetase (metRS); to our knowledge there is no aminoacyl-tRNA synthetase (aaRS) inhibitor in clinical anti-malarial development, and structural differences between several *Pf*aaRS and their human counterparts leading to selective *Pf*aaRS inhibitors are encouraging[43].

The involvement of protein and lipid kinases in key pathogen functions have made inhibitors thereof a focus of drug design strategies including those that affect multiple life cycle stages[44]. Among the asexual stage active compounds, MMV1580482 (URMC-099) operates as a human MLK3 inhibitor and MMV1593539 as a *Staphylococcus aureus* pyruvate kinase (PK1) inhibitor, both of interest as kinase targets with a crystal structure for PK1 are available (10.2210/pdb3KHD/pdb), which could guide selectivity and optimization studies, if validated as the target.

Importantly, our parallel screening approach on different life cycle stages yielded compounds and chemical scaffolds that not only have stage-specific asexual parasite activity but also selectively and specifically target the elusive gametocyte stages with activity in mosquito transmission assays. This unbiased approach,

in place of the paradigm where compounds are only profiled for additional life cycle activity once asexual activity has been established, confirms the possibility of identifying gametocyte-specific compounds[7,8,10]. Indeed, we identified several active compounds that have no former documented antimalarial activity, simply because they were previously not screened against the correct life cycle stage of the parasite, where the relevant biology being targeted was essential.

Our stringent profiling cascade additionally ensured a high success rate in confirming TBA and validates the use of orthogonal gametocytocidal screens[4,10] as a primary filter in large-scale screens. In addition, a linear correlation between gametocytocidal and male gamete activity was present, which translated to oocyst reduction. By evaluating both TRA and TBA, our data highlighted the importance of both parameters in evaluating SMFA data. Notably, we showed a large reduction in TRA for some compounds; the decreased number of oocysts carried by treated mosquitoes will reduce transmission, as mosquito infection intensity is critically important to the success of transmission[45]. However, the additional decrease in oocyst prevalence (TBA) implies that the majority of treated mosquitoes would not carry parasites, which could have an epidemiological impact, in line with World Health Organization-recommended vector control interventions.

Gametocyte-targeted compounds would presumably target divergent biological processes compared to asexual parasites[46], and this, in addition to the low parasite numbers in transmission stages and non-proliferative nature of gametocytes, would reduce the risk of resistance development. When used in combination with a TCP-1 candidate, such TCP-5 targeted compounds could protect the TCP-1 drug from resistance development. Alternatively, when used as a stand-alone drug, TCP-5 targeted compounds could decrease the gametocyte burden in the human population, which would be particularly important in pre-elimination settings as add-ons to enhance standard measures of malaria control.

Our data indicate specific gametocyte-associated biological processes worthy of further investigation, including two imidazoles (eberconazole, MMV1634492 and MMV1634491). The antimalarial activity of eberconazole, a fungal ergosterol biosynthesis inhibitor[47], has not been described before and may point to an unexplored mode of action. We also identified ML324 with selective activity against early- and late-stage gametocytes, similar to recent reports against immature gametocytes[29]. As a known PfJmj3 inhibitor, we show that ML324 potently kills male gametes with confirmed TBA and does so by preventing histone demethylation, at least for H3K9me as heterochromatin mark 3 in gametocytes[30]. As a consequence, crucial genes required for the maintenance of chromatin dynamics during gametocyte development are affected and a heterochromatic, silenced gene state is maintained, preventing gamete formation. The distinct change in the transcriptional programmes agrees with the effect of jumonji histone demethylase inhibitors in Plasmodium spp.[29], but contrast the effect seen with histone methyltransferase[31] or the more global changes of up to 60% of the transcriptome seen with histone deacetylase inhibitors[48]. Gametocytes and male gametes are therefore highly sensitive to changes in histone methylation due to ML324 treatment, similar to what is seen for other Jmj demethylase inhibitors[29].

Lastly, our screens identified two compounds that are established inhibitors of MmpL3 in bacteria[19–21]. Albeit structurally dissimilar, both compounds inhibit MmpL3 through interaction with the protein pore section as indicated by co-crystallization data[49]. A homologue for this protein is not detectable in the Plasmodium genome, and our mechanistic data show that neither is the PfNCR1 as MmpL3-family member a target, implying a non-MmpL3-associated mechanism of action. Rather, SQ109 pressure generates mutations in the PfVapA. The possibility that this protein is not the direct target of SQ109 and only involved in a SQ109 resistance mechanism cannot be excluded without further validation. However, if confirmed as target, PfVapA could present a druggable target in Plasmodium as has been implicated for another antimalarial candidate class, triaminopyrimidine (TAP)[50], and could be explored[51] similar to bedaquiline as antitubercular targeting the F-type ATPase[52]. The SQ109 mutations are within the catalytic site, whereas those for TAP are not. The more potent action of SQ109 on the non-proliferative differentiated P. falciparum gametocytes may indicate similar action as for the non-proliferative and transmissible trypomastigotes of Trypanosoma cruzi[32]. MmpL3 is also absent in Trypanosoma and Leishmania[53] and SQ109's activity is pleiotropic and includes disruption of proton motor forces across membranes as an uncoupler[32]. Such additional ionophore effects is reminiscent again of additional actions seen for bedaquiline[54]. These mechanisms are currently being investigated for P. falciparum, including possible disruption of mitochondrial respiration[46,55] and lipid metabolism/transport, which is essential to gametocytogenesis[56] and oocyst development[57].

Our data therefore provide an extensive data set of compounds that could be repurposed or redirected for antimalarial development. One advantage of screening biologically active screening collections such as the PRB is that they contain a large number of compounds with well-described DMPK profiles and empirically determined physical characteristics and that they could progress rapidly through the drug discovery pipeline. Importantly, front-loading screening efforts with biologically active compounds also point towards proteins that can be targeted in multiple parasite life cycle stages.

## Methods

**Ethics statement.** This work holds ethical approval from the University of Pretoria Health Sciences Ethics Committee (506/2018), University of Cape Town: AEC017/026, University of the Witwatersrand Human Research Ethics Committee (M130569) and Animal Ethics Committee (20190701-7O), CSIR Research Ethics Committee (Ref 10/2011) and Scripps Research's Normal Blood Donor Service (NBDS), with approval under IRB Number 125933.

**Parasite culturing.** P. falciparum asexual parasites were cultured in vitro from drug-sensitive strain NF54 (PfNF54), drug-resistant strain Dd2 (PfDd2, chloroquine, pyrimethamine and mefloquine resistant) and the luciferase reporter line NF54-Pfs16-GFP-Luc (kind gift from David Fidock, Columbia University, USA)[58] were used for the various downstream analyses. Gametocytogenesis was induced from asexual NF54-background parasites as described[4].

**Asexual blood stage screening**

*Parasite lactate dehydrogenase (pLDH) assay.* The asexual blood stage antiplasmodial activity was assessed on NF54 P. falciparum using the pLDH assay[59,60]. Initially, all compounds were screened on two separate occasions by seeding ring-stage cultures (1% haematocrit, 2% parasitaemia) in 96-well plates and adding compounds at two concentrations only (2 and 20 μM), and survival was determined after 72 h under an appropriate atmosphere (4% $CO_2$, 3% $O_2$ and 93% $N_2$) at 37 °C, before survival was determined colorimetrically at 620 nm. The antiplasmodial $IC_{50}$ was determined for suitably active compounds under the same culture conditions as the single-point screens. Chloroquine and artesunate were used as control compounds for all the antiplasmodial assays.

*SYBR green I assay.* Antiplasmodial activity on drug-resistant Dd2 P. falciparum was determined with SYBR Green I fluorescence[13]. Briefly, parasite suspension (0.3% parasitaemia, 2.5% haematocrit) was dispensed into 1536-well black, clear bottom plates with pre-spotted compounds using a MultiFloTM Microplate dispenser (BioTek) at a volume of 8 μL/well. The plates were incubated at 37 °C for 72 h in a low-oxygen atmosphere, after which 10× SYBR Green I (Invitrogen) in Lysis buffer (20 mM Tris/HCl, 5 mM EDTA, 0.16% (w/v) saponin, 1.6% (v/v) Triton X) was added using MultiFloTM Microplate dispenser (BioTek) at a volume of 2 μL/well. The plates were incubated in the dark at room temperature (RT) for 24 h. P. falciparum proliferation was assessed by measuring the fluorescence from the bottom of the plates by using the EnVision® Multilabel Reader (PerkinElmer) (485 nm excitation, 530 nm emission). $IC_{50}$ values were determined in CDD vault

(https://www.collaborativedrug.com/) normalized to maximum and minimum inhibition levels for the positive (Artemisinin, 5 µM) and negative (dimethyl sulfoxide (DMSO)) control wells.

### Gametocyte screening[4]

*PrestoBlue® fluorescence assay.* Stage IV/V gametocyte cultures (*Pf*NF54, 2% gametocytaemia, 5% haematocrit, 100 µL/well) were seeded with compounds (in DMSO) and incubated at 37 °C for 48 h under hypoxic conditions, stationary, after which 10 µL of PrestoBlue® reagent was added to each well and incubated at 37 °C for 2 h. Supernatant (70 µL) was transferred to a clean 96-well plate and fluorescence was detected at 612 nm. Dihydroartemisinin (DHA) was used as positive kill control.

*ATP bioluminescence assay.* Stage IV/V *Pf*NF54 gametocyte cultures were enriched with density gradients[4] and 30,000 gametocytes were seeded into 96-well plates in the presence of compound and incubated for 24 h at 37 °C. ATP levels were determined with a Promega BacTiter Glo Bioluminescence system[4]. DHA or methylene blue (MB) were used as positive kill control.

*Luciferase reporter assay.* Stage IV/V gametocytes were produced from the NF54-*pfs16*-GFP-Luc line[58] seeded at 2% gametocytaemia and 2% haematocrit with compounds (in DMSO) and incubated for 48 h under hypoxic conditions (stationary) at 37 °C. Luciferase activity was determined in 30 µL parasite lysates by adding 30 µL luciferin substrate (Promega Luciferase Assay System) at RT and detection of resultant bioluminescence at an integration constant of 10 s was performed with a GloMax®-Multi+ Detection System with Instinct® Software[4]. DHA, MB and MMV390048[61] were used as positive drug controls and $IC_{50}$ was determined with non-linear curve fitting (GraphPad Prism 6) normalized to maximum and minimum inhibition (DMSO control wells).

### *P. berghei* liver stage assay[13]

Compounds with potential causal prophylactic activity were tested in HepG2-A16-CD81 cells seeded in 1536-well plates (Greiner Bio) containing 50 nL of test and control compounds diluted in DMSO and incubated for 24 h. Thereafter, *P. berghei* sporozoites (*P. berghei* ANKA GFP-Luc-SMcon) freshly obtained by dissecting salivary glands of infected *A. stephensi* mosquitoes were added to each well at a density of $1 \times 10^3$ per well. The plates were centrifuged for 5 min at $330 \times g$ and incubated at 37 °C. Forty-eight hours post infection, 2 µL of luciferin reagent (Promega BrightGlo) was added to each well, and luciferase activity was detected using a Perkin Elmer Envision plate reader. $IC_{50}$ values were determined in CDD vault (https://www.collaborativedrug.com/) normalized to maximum and minimum inhibition levels for the positive (GNF179, 5 µM) and negative (DMSO) control wells.

### Cytotoxicity counter-screening

*CHO toxicity screening.* General cytotoxicity of the compounds was determined using the 3-(4,5-dimethyl-thiazol-2-yl)-2,5-diphenyltetrazolium bromide assay[62] by exposing CHO cells for 48 h at 2 and 20 µM of each compound, and survival was determined colorimetrically. Cells were seeded at a density of $10^6$ cells/well in 200 µL of suitable medium and left overnight (O/N) to attach. After 24 h, medium was gently aspirated from the culture plates, replaced with 200 µL of fresh medium containing the compounds and incubated for a further 48 h. Emetine was used as the control compound for cytotoxicity assessment.

*HepG2 toxicity screening.* Hepatocellular carcinoma line (HepG2-A16-CD81) was seeded as above for the sporozoite liver stage assay but in the absence of sporozoites. HepG2 toxicity measurements was performed by adding Promega Cell-TiterGlo® (2 µL) followed by luminescence measurement. Curve fitting was done as described above using puromycin (25 µM) as a positive control and DMSO as negative control wells.

### Chemical clustering

Drug classes and biological pathways/protein targets were identified based on text and structure searches (PubChem, DrugBank and Sci-Finder), and chemical space analysis was performed with StarDrop v 6.6 (https://www.optibrium.com/stardrop/) based on structure similarities. The launched drug space was generated from the data file available within the Stardrop software. The antimalarial drug space was generated using marketed antimalarial drugs and compounds undergoing clinical trials. The connectivity network was constructed by clustering the compounds using the *FragFP* descriptor (Tanimoto similarity index >0.50) in OSIRIS DataWarrior v 5.0.0 (www.openmolecules.org).

### Male gamete exflagellation inhibition assay (EIA)[63]

The EIA was performed as described[63] on >95% stage V gametocytes, resuspended in 30 µL of ookinete medium (culture media supplemented with 100 µM xanthurenic acid and 20% (v/v) human serum, A+ male). For the carry-over experiment, mature stage V gametocytes were treated with 2 µM of each compound for 48 h prior to induction of exflagellation in the presence of compound. In the washout format, the same was done but drug washed out before exflagellation was induced. In the direct format,

drug (2 µM) was only added during induction of exflagellation in the ookinete medium. Exflagellating centres were evaluated on 10 µL activated culture settled in a Neubauer chamber at RT. Movement was recorded by video microscopy (Carl Zeiss NT 6V/10W Stab microscope with a MicroCapture camera, ×10 magnification) and semi-automatically quantified from 15 videos of 8–10 s each (captured at random locations) between 15 and 22.5 min after incubation.

### Mosquito rearing and SMFA

*A. coluzzii* mosquitoes (G3 colony, species confirmed in refs. [64,65]) were reared under biosafety level 2 insectary conditions (80% humidity, 25 °C, 12 h day/night cycle with 45 min dusk/dawn transitions[66]) on a 10% (w/v) sucrose solution diet supplemented with 0.05% 4-aminobenzoic acid. SMFA was conducted using glass feeders (covered with cow intestine/sausage casing to form a membrane) on top of feeding cups (350 mL), maintained at 37 °C. A mature stage V gametocyte culture (1.5–2.5% gametocytaemia, 50% haematocrit in A+ male serum with fresh erythrocytes) was functionally evaluated for male gamete exflagellation and male:female ratio of 1:3 confirmed before proceeding with feeding. The gametocyte culture was either untreated or treated with 2 µM of each compound 48 h. Per glass feeder, 1 mL of the gametocyte culture was added. Each cup contained 25 unfed (2–3 h starvation) *A. coluzzii* females (5–7 days old), which were allowed to feed in the dark for 40 min. All unfed or partially fed mosquitoes were removed, and remaining females were housed as above for 8–10 days. Mosquitoes were dissected to remove midguts, which were rinsed in phosphate-buffered saline (PBS) and placed into 0.1% (v/v) mercurochrome for 8–10 min and oocysts counted under bright field illumination at ×20–×40 magnification. The percentage of block in transmission (reduction in prevalence) and percentage of reduction in number of oocysts (intensity) were calculated after normalizing to the control untreated sample. TBA (%TBA: $\frac{Cp-Tp}{Cp}*100$, where

*p*: oocyst prevalence, *C*: control and *T*: treated) and TRA (%TRA: $\frac{Ci-Ti}{Ci}*100$, where *i*: oocyst number (intensity), *C*: control and *T*: treated)[67] were determined. Each biological replicated experiment included two technical repeats (two feeding cups per compound), and this was repeated for at least three independent biological experiments per compound. Non-parametric *T* test (Mann–Whitney) was applied (GraphPad Prism 8.3.0) for statistical analysis.

### Endectocide evaluation

Selected compounds were evaluated for endectocidal activity by using SMFA as above but with 2 µM of each compound in cow blood (100 µL) at 37 °C for 35 min feeds. Each 350 mL feeding cup contained 30, 4 h starved and 2–4-day-old *A. coluzzii* females. Fully fed or partially fed females were retained for daily monitoring of mortality under standard insectary rearing conditions, for up to 4 days post treatment. Ivermectin (1 µM) (Virbac, SA) and DMSO (0.02% (v/v)) was used as controls. Between three and five independent replicates were performed per compound.

### Histone methylation detection

Late-stage gametocytes were treated with ML324 (5 µM) for 24 h after which histones were extracted from gametocytes for dot blots and enriched as described before[68]. Sampled gametocytes were released from host erythrocytes using 0.06% (w/v) saponin in PBS. The isolated gametocytes were washed in PBS to remove residual erythrocyte debris. Nuclei were isolated using a hypotonic lysis buffer (10 mM Tris HCl, pH 8.0, 3 mM $MgCl_2$, 0.25 M sucrose, 0.2% (v/v) Nonidet P-40) with a protease inhibitor (PI) cocktail and then collected by centrifugation for 10 min at $500 \times g$ and 4 °C. This step was repeated and nuclei were washed in the same buffer lacking Nonidet P-40 and finally pelleted nuclei were resuspended in a 10 mM Tris buffer (pH 8.0) containing 0.8 M NaCl, 1 mM EDTA and a PI cocktail. Resuspended nuclei were incubated on ice for 10 min and then collected through centrifugation at $500 \times g$ for 5 min at 4 °C. Histones were isolated by acid extraction in 0.25 M HCl with rotation at 4 °C O/N. The acid-soluble protein-containing supernatant was retained after centrifugation at $11,000 \times g$ for 30 min. This was mixed with an equal volume of 20% (v/v) trichloroacetic acid and incubated on ice for 15 min, with the pellet collected by centrifugation at $12,000 \times g$. The pellet was washed in acetone, collected again at $12,000 \times g$ for 15 min and then reconstituted in ddd$H_2O$. The histone-enriched supernatant was retained following centrifugation at $5000 \times g$ for 2 min. The isolated histones were quantitatively spotted onto nitrocellulose membranes in technical triplicate. The membranes were blocked in TBS-T (50 mM Tris pH 7.5, 150 mM NaCl, 0.1% (v/v) Tween) containing 5% (w/v) blotting-grade blocker (Bio-Rad) for 1 h and probed O/N with anti-H3K9me3[30] antibody (Abcam ab8898, 1:10,000), anti-H3K9ac[68] (Abcam ab4441, 1:10,000) and anti-H3 core[30] (Abcam ab1791, 1:10,000) with antibody dilutions prepared in TBS-T. Membranes were then incubated in goat anti-rabbit horseradish peroxidase-conjugated secondary antibody (Abcam ab6721, 1:10,000) and visualized using Pierce SuperSignal West Pico PLUS Chemiluminescent Substrate. Relative abundance units ($n = 3$, ±S.E.) for paired treated and vehicle control samples were calculated using Image J 1.53a.

### DNA microarrays of ML324-treated parasites

DNA microarrays (60-mer, Agilent Technologies, USA) based on the full *P. falciparum* genome[69] that included 5441 annotated transcripts[46,70], were performed as before[46]. *P. falciparum* NF54 (1–3% gametocytaemia, 4% haematocrit in replicates, stage II/III gametocytes) was

treated with 5 µM ML324 (Sigma-Aldrich, SA) for 24 h. Gametocytes were isolated with 0.01% (w/v) saponin and total RNA was isolated using a combination of TRIzol (Sigma-Aldrich, USA) treatment and phenol–chloroform extraction as described before[46]. In all, 2.5–8 µg of the RNA was used to synthesize cDNA for each sample (untreated and ML324-treated gametocytes) and dye-coupled to Cy5 (GE Healthcare, USA) and hybridized to the array with an equal amount (≥350 ng) of Cy3-labelled (GE Healthcare, USA) reference cDNA (reference pool containing cDNA from each gametocyte sample and mixed stage 3D7 asexual parasites). The slides were scanned (5 µm resolution, at wavelengths of 532 nm and 633 nm for Cy3 and Cy5, respectively) using an Agilent SureScan G2600D scanner. Agilent Feature Extractor Software (v 11.5.1.1) was used to extract normalized signal intensities according to the GE2_1100_Jul11_no_spikein protocol[69]. Robust-spline within-slide and G-quantile between-slide normalization of array data was performed using the limma and marray packages in R (v3.2.3, www.r-project.org) with $\log_2$-transformed expression values ($\log_2$ Cy5/Cy3) obtained from the fitted linear model. Genes with $\log_2$-transformed FCs ($\log_2$ FC) ≥ 0.5 in either direction were defined as differentially expressed in treated gametocytes. Differentially expressed genes were visualized using TIGR MeV (v4.9.0, www.tm4.org) and functionally classified by gene ontology (GO) enrichment analyses (biological processes, $p$ values ≤ 0.05) in PlasmoDB (v46, www.plasmodb.org). The full data set is available in GEO with accession number GSE157420. GO analysis was performed with the GO enrichment tool in PlasmoDB with a $p$ value cut-off of 0.01 on the level of biological process.

**PfNCR1 cross-resistance analysis.** Concentration–response curves with MMV687273 (SQ109) were determined using a parasite clone in which the expression of PfNCR1 is regulated by anhydrotetracycline (aTc)[33]. aTc (500 nM) was removed by washing synchronous young ring-stage parasites three times for 10 min in culture media. To a portion of the parasite culture, 500 nM aTc was added back. Parasites with 500 nM aTc or without aTc were diluted to a parasitaemia of 1.1% and incubated with compounds diluted in DMSO. The concentration of DMSO was kept constant at 0.1% (v/v). The PfNCR1-specific compound MMV009108 was used as a positive control. Parasitaemias of acridine orange-stained parasites were measured on a BD FACS Canto after 72 h and in technical triplicate for each compound dilution. Data were processed using GraphPad Prism 8.3.0 and fit with nonlinear regression analysis.

**Mutant generation and whole-genome sequencing and analysis of SQ109-resistant parasites.** P. falciparum clonal line 3D7-A10 (Saint Louis)[71] was cultured at 2% haematocrit in O+ erythrocytes in RPMI supplemented with 0.5% (w/v) Albumax II. Cultures were grown statically in 5% $O_2$, 5% $CO_2$ and 90% $N_2$. To generate resistant parasites, 3D7-A10 cultures were exposed to increasing concentrations of SQ109. Parasites that returned after selection were cloned by limiting dilution. Growth inhibition assays were performed in 200 µL volumes and growth was quantified using Pico Green (at a 1:200 dilution; Life Technologies). Pico Green fluorescence was quantified using the fluorescein filter set on an EnVision Multilabel Platereader (Perkin Elmer). Growth assays commenced with synchronized ring-stage parasites at 0.5% parasitaemia and growth was measured after 72 h. $EC_{50}$ values were calculated by fitting a non-linear curve using the expression Parasite Proliferation = $y_{min} + y_{max}/(1 + [(\text{drug concentration})/EC_{50}]$ $b$), where $y$ is parasite proliferation and $b$ is a fitted constant.

Sequencing libraries were prepared from DNA extracted from the mutant and parental cell lines with the Nextera XT Kit (Illumina), using the standard dual index protocol. These were sequenced on the Illumina HiSeq 2500 in Rapid Run mode, with an average read length of 100 base pairs. Reads were aligned to the P. falciparum 3D7 reference genome (PlasmoDB v. 13.0)[51], with single-nucleotide variants (SNVs) and insertion/deletions called with the Genome Analysis Toolkit's HaplotypeCaller[72–74]. SNVs were removed if they met the following criteria: ReadPosRankSum >8.0 or <−8.0, QUAL < 500, Quality by Depth <2, Mapping Quality Rank Sum <−12.5, and filtered depth <7. Mutations where read coverage was <5 and/or where mixed-read ratios were >0.2 (reference/total reads) across all samples were removed. Variants were annotated using SnpEff[75]. Variants present in the resistant clones but not in the parent clone were identified. Since all parasite lines were cloned before sequencing, only homozygous variant calls were retained. Sequencing files were deposited in the National Center for Biotechnology Information Sequence Read Archive database with accession code PRJNA659232. Mutations in the V-type H+-ATPase subunit A (PF3D7_1311900) were confirmed by direct sequencing.

**V-type H+-ATPase subunit A modelling.** The V-type H+-ATPase subunit A (PF3D7_1311900) was modelled onto a complete structure of mammalian V-type H+-ATPase from rat brain (pdb code 6vq6.1) and is available in the Swiss-Model Repository[76] with access code Q76NM6. Structure assessment was performed with Ramachandran and MolProbity. Protein characteristics were evaluated with UniProt: https://www.uniprot.org/uniprot/Q76NM6, and InterPro: https://www.ebi.ac.uk/interpro/protein/UniProt/Q76NM6/. Structure fitting and visualization was performed with UCSF Chimera (https://www.cgl.ucsf.edu/chimera/)[77].

**Data analysis.** Assay platforms' reproducibility was evaluated by Z′-factors (Supplementary Data 1), and data are from a minimum of three independent biological repeats as indicated. Statistical evaluation was performed with paired, two-tailed $t$ test for most assays, non-parametric $t$ test (Mann–Whitney) was applied for the SMFA data (GraphPad Prism 8.3.0) and Fisher's exact two-tailed test for GO annotations. Supra-hexagonal maps were generated in Rstudio 1.1.456 with the RColorBrewer R package.

**Reporting summary.** Further information on research design is available in the Nature Research Reporting Summary linked to this article.

## Data availability

The authors declare that all relevant data supporting the findings of this study are available within the paper and provided as Supplementary Data files (Files 1–4), supplementary figures (Figs. 1–8) and supplementary tables (Tables 1–3) are included. Transcriptome data is available from GEO (accession code GSE157420), whole-genome sequencing data from NCBI Sequence Read Archive (PRJNA659232) and structural data from Swiss Model Repository (Q76NM6). Source data are provided with this paper.

## Code availability

Codes used to analyse data are available at GitHub (https://github.com/Ash-bot/Beehive-inhibition-plots).

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

## Acknowledgements

We thank the MMV for assembly and supply of the PRB and Didier Leroy and Esperanza Herreros from the MMV for helpful discussions. We thank Annette Bennett for prior optimization of the SMFA. We thank the Medicines for Malaria Venture and South African Technology Innovation Agency (TIA) for funding (Project MMV18/0001). This project was in part supported by the South African Medical Research Council with funds received from the South African Department of Science and Innovation, in partnership with the Medicines for Malaria Venture (KC, LMB, LLK and TLC); and the DST/NRF South African Research Chairs Initiative Grant (LMB UID: 84627, LLK UID: 171215294399; KC UID: 64767); and CSIR Parliamentary Grant funding (AT, DM). EAW and EI thanks the Bill and Melinda Gates Foundation for funding (OPP1054480 and OPP 1054480) and NS was funded by the Australian NHMRC (APP1072217).

## Author contributions

JD, LMB, KC, LLK, GB, EAW and TLC conceptualized the work and supervised the data acquisition. JR, MvdW, DT, NM, SO, AT, PM, SL, BB, EE, NV, LN, AC, NS, JC, DO, EI, LMO performed experiments and data analysis with supervision from KC, EAW, DM, DG, ML, LLK, TLC. CLM, AvH, AH, GB, DC, LLK and LMB performed additional data analysis. LMB, MvW and JR wrote the manuscript with contributions from all authors.

## Competing interests

The authors declare no competing interests.
