## [Peer Review File · Nature Communications]

Reviewers' comments:

Reviewer #1 (Remarks to the Author):

New antimalarials are urgently needed, and antimalarials with activity against multiple parasite stages are especially valued for the potential to block infection or transmission. In this work, the authors screen the MMV Pandemic box (400 drug-like compounds) in parallel against a series of Plasmodium life stages and for mosquito activity. Gametocyte hits were confirmed using 3 orthologous assays. Hits were defined as modest activity (>50% inhibition at 2 microM of asexual parasites or 5 microM gams/liver). Only 23 of 72 hits with asexual activity had confirmed IC50s under 2 microM (line 159) against the same strain. In addition to hits with multistage, asexual, or gametocyte-targeting potential, they identify 4 distinct chemotypes that target gametocytes but that had not previously been recognized for their antimalarial activity. Two compounds (birinapant and AZD-0156) have pan-stage activity against Plasmodium.

Overall, this is a clearly written manuscript. The topic of antimalarial drug discovery is compelling and novel pan-stage hits will be of interest. However, there are a number of concerns with the manuscript. The rigor of hit selection and downstream assays is not clear from the data presented, and the high rate of false positives in the initial screen is concerning. Mechanistic data is not provided for any compound class, so the primary impact of the findings is the new chemical matter and new targets are not validated. One of the top hits, birinapant, has been previously described as an inhibitor of liver-stage malaria, decreasing the novelty of this finding while raising questions as to why the prior study did not identify asexual activity. In addition, the parasite selectivity of this compound is potentially insufficient (possible 5-fold, although CC50 against HepG2 cells not completely determined). Additional concerns are noted with respect to replicates and error.

Major critiques:

- 1) It seems somewhat surprising that there were so many false positive hits for asexual activity (~30% were true positives on rescreening). This suggests that investigators should have used statistical cut off (rather than efficacy cutoffs) to define a hit.
- 2) Related to (1) above: measures of assay reproducibility (Z factor) should be provided for all screening assays.
- 3) Fig 1B: this distribution of hits seems based on the hits, not the validated hits. Since the false positive hit rate was high, these distributions are somewhat misleading. A similar concern could be made for the ABS
- 4) Fig. 3: error measurements or confidence intervals should be provided for all assay measurements
- 5) Fig. 5 B, C, D. Please clarify that all assays were performed at least 3 independent times. Assay variability should be clearly indicated.
- 6) Fig 6. All assays should be performed a minimum of 3 times. Error measurements should be provided (not visible on all bars).
- 7) Line 178: since no target validation was performed, conclusions re: DHFR in gametocytes is premature based on the provided data alone.

Reviewer #2 (Remarks to the Author):

The need for novel scaffolds targeting multiple plasmodial life cycle stages is a high priority in the backdrop of resistance developing against front line antimalarials.

The manuscript presents a well-designed parallel screening of the MMV-PRB compounds for potential antimalarial activity against asexual PfNF54, stage IV/V gametocytes from PfNF54 and *P. berghei* liver stages. The data is presented in a logical and coherent manner and the study design has built in successful validations to corroborate the results presented.

Questions:

-While the supplementary data does provide detailed info for the full screen, it would be good to add in a 'filtered file' containing the selection criteria and relevant data for the post-cytotoxicity filtered panel. It is difficult to ascertain on which compounds the CC50 (HepG2) and viability (CHO) assays were done and the corresponding selectivity indices. Perhaps a more focussed presentation of this data in the manuscript will enable the reader to grasp the major finding of the study more readily.

-Figure 6 is quite complex and difficult to follow. Could the authors consider an alternate mode of presentation of this data (preferably simplified).

-The detection of the two non-cytotoxic multistage PRB compounds (Birinapant and AZD-0156) and a range of dual active compounds is indeed exciting. However, the discussion then goes on to speculate potential hurdles for the progression of these leads (discrepancies with previous in vivo studies, in vitro cytotoxicity etc.). The value of this very well carried out study would probably be best realised if in-vivo screens and data on PK/PD and Tox are carried for the selected candidates with atleast dual stage effectivity. Having said the, the vast amount of info contained in the manuscript will most certainly be beneficial if released into the wide research community and the manuscript merits publication, albeit in a revised format.

Reviewer #3 (Remarks to the Author):

The Pandemic response box, studied in the present manuscript, is one of the "boxes" made freely available to researchers within the Open Source Drug Discovery Programme of Medicine for Malaria Venture (<https://www.mmv.org/mmv-open/open-source-drug-discovery-programme>) to speed up drug research on malaria, then extended to Neglected Tropical Diseases and lately to bacteria, virus and fungi.

As reported in the MMV website, the first MMV Malaria Box was launched in 2011 and since then 58 related papers have been published in peer review journals. The MMV Pathogen Box was launched in 2015 and since then 38 related papers have been published in peer review journals. The results of such a research effort are freely available since they are placed in open access public repository such as ChEMBL (<https://www.ebi.ac.uk/chembl/>)

The Pandemic response box (intriguing name at present) was launched on Jan 2019 and there is only one related paper in Pub Med up to June 2020.

The present manuscript by Janette Reader et al. entitled "Multistage and transmission-blocking targeted antimalarials discovered from the open source MMV Pandemic Response Box" reports a considerable amount of work for the in vitro screening the Pandemic Response Box against several stages of the Malaria Parasite, *P. falciparum*, with the intent to identify new lead compounds.

The work is well done, the methodology is appropriate. There are two positive aspects in the

manuscript

1- parallel screening of the same compounds against different stages (asexual parasites, stage IV/V gametocytes, gametes, oocysts and liver stages) of *P. falciparum* and for endectocidal activity
2- the application of orthogonal assays against gametocytes which validate the results.

Overall, the data are interesting since they contribute to explore different chemical spaces as sources of novel antimalarials, but also relatively disappointing since none of the active compounds shows an IC50 in the low nanomolar range, as expected for a promising lead, at least against the asexual stage. The only one was MMV1580844 (IC50 0.0017 μM) a DHFR inhibitor, very active also against *P. berghei* liver stages, which unfortunately showed strong cross resistance with pyrimethamine.

The most interesting compounds are those specifically and exclusively targeting stage IV-V gametocytes. Two of the most active, namely MMV1580843, MMV687273, have a known target (MmpL3) in bacteria, but their mode of action in *P. falciparum* has not been elucidated, yet. A deeper investigation of some of the most active compounds would have made the manuscript more innovative. As it is, the work appears largely descriptive, similar in the approach to the above mentioned work in public domain. Therefore, in my opinion, the present manuscript is not sufficiently innovative to be within the scope of this journal which aims to publish "...important advances of significance to specialists within each field"

Multistage-active and transmission-blocking targeted antimalarials discovered from the open-source MMV Pandemic Response Box

Janette Reader^{1a}, Mariëtte E. van der Watt^{1a}, Dale Taylor², Claire Le Manach², Nimisha Mittal³, Sabine Otilie³, Anjo Theron⁴, Phanankosi Moyo¹, Erica Erlank⁵, Luisa Nardini⁵, Nelius Venter⁵, Sonja Lauterbach⁶, Belinda Bezuidenhout⁶, Andre Horatscheck², Ashleigh van Heerden¹, Natalie J. Spillman⁷, Anne N. Cowell³, Jessica Connacher¹, Daniel Opperman¹, Lindsey M. Orchard⁸, Manuel Llinás^{8,9}, Eva S. Istvan⁷, Daniel E. Goldberg⁷, Grant A. Boyle², David Calvo¹⁰, Dalu Mancama⁴, Theresa L. Coetzer⁶, Elizabeth A. Winzeler³, James Duffy¹¹, Lizette L. Koekemoer⁵, Gregory Basarab², Kelly Chibale^{2,12}, Lyn-Marié Birkholtz¹

Point-by-point responses to reviewers comments:

REVIEWER 1

Overall, this is a clearly written manuscript. The topic of antimalarial drug discovery is compelling and novel pan-stage hits will be of interest. However, there are a number of concerns with the manuscript. The rigor of hit selection and downstream assays is not clear from the data presented, and the high rate of false positives in the initial screen is concerning.

Response: We apologise if there may have been some confusion regarding downstream validation of the hits obtained. We clarified in the text that all the hits from the primary screens were validated in downstream assays, and from these, only the most potent subset was further evaluated. We additionally provide statistical validation of our hit selection criteria as well as parameters for evaluation of all our assay platforms (see below).

Mechanistic data is not provided for any compound class, so the primary impact of the findings is the new chemical matter and new targets are not validated. One of the top hits, birinapant, has been previously described as an inhibitor of liver-stage malaria, decreasing the novelty of this finding while raising questions as to why the prior study did not identify asexual activity. In addition, the parasite selectivity of this compound is potentially insufficient (possible 5-fold, although CC50 against HepG2 cells not completely determined).

Response: We agree that the primary finding is new chemical matter but as we state in the discussion, this is however a unique and potentially powerful dataset, as it explores a wide range of chemical matter that has already been well worked up in preclinical (and clinical) studies and has the potential to be repurposed or redirected in optimisation programmes. We do believe that, not only should our data have an immense impact in the malaria drug discovery community (a fact already evident in the number of requests for information based on the pre-print release of this paper), but our data also provides a comprehensive starting point to explore new biological pathways and regulatory mechanisms.

In addition to the above, the reviewer critiques the lack of mechanistic data in the previous version of the paper. To extend the innovation of the paper and address the reviewer's concern, we have therefore now included mechanistic data on two very interesting compounds: ML324, the most potent transmission-blocking antimalarial; and SQ109, a compound far advanced as antitubercular clinical candidate. This is the first time these two compounds were explored in this manner, particularly in gametocytes, and we believe that our findings are highly significant and novel, and will dramatically increase the impact of the work in the field. We now describe two novel potential drug targets in gametocytes, a histone demethylase and the V-type H⁺-ATPase. We include two new figures (Figure 6 and 7) to explain these findings.

Regarding the comments on Birinapant, we agree that a moderate selectivity towards the parasite was obtained. However, such data from primary screens does not preclude further development of such compounds in medicinal chemistry programs; indeed, the majority of preclinical candidates currently in development had some toxicity or solubility issues that were fixed as part of optimisation strategies. Moreover, a large number of other chemotypes described in the paper are highly selective to the parasite in our dataset and of interest for further development against different life cycle stages, and data on one compound should not preclude publication of the rest of the findings. The question as to why Birinapant was not previously identified with asexual activity can simply be ascribed to evolution of screening methods, different species/strains of parasites used and screened with different assay platforms. The fact remains that we picked up this compound with potent asexual activity against two different *P. falciparum* strains on two different assay platforms, providing confidence in our data.

Major critiques:

1) *It seems somewhat surprising that there were so many false positive hits for asexual activity (~30% were true positives on rescreening). This suggests that investigators should have used statistical cut off (rather than efficacy cutoffs) to define a hit.*

Response: As stated above, we apologise if there may have been some confusion regarding downstream validation of the hits obtained in our original text. All the hits from the primary screens retained activity in downstream assays. We applied an additional stringent progression threshold on these confirmed hits and only the most potent subset (IC₅₀ less than 2 µM), leading to the '30%' with that was progressed. We have now hopefully clarified this in the text. Additionally, as requested by the reviewer, we applied a statistical cut-off at 3 S.D.s around the mean for all the screens and we confirm that our hits fall within these statistical criteria. This information is also now included in the text.

2) *Related to (1) above: measures of assay reproducibility (Z factor) should be provided for all screening assays.*

Response: Z' factors were included for all assays as requested as part of a supplementary table.

3) *Fig 1B: this distribution of hits seems based on the hits, not the validated hits. Since the false positive hit rate was high, these distributions are somewhat misleading. A similar concern could be made for the ABS*

Response: Again, as clarified above, the primary screens did not have high false positive rates and all hits were validated on rescreening. We therefore did not change this figure, as it does give an accurate depiction of the spread of hits obtained for all the compounds in the PRB.

4) *Fig. 3: error measurements or confidence intervals should be provided for all assay measurements*

Response: We did include error measurements for all assays used in this figure in the form of S.E. Additionally, we now also included 95% confidence interval data on all IC₅₀ values in this figure as supplementary data.

5) *Fig. 5 B, C, D. Please clarify that all assays were performed at least 3 independent times. Assay variability should be clearly indicated.*

Response: Figures 5 B, C, and D were for at least 3 independent experiments and variability was indicated with S.E. In light of data transparency, individual data points were now included on histograms around the mean (±S.E.) values and 95% confidence interval spread was included on all datapoints on any IC₅₀ curves. These figures were now moved to new Figure 7.

6) *Fig 6. All assays should be performed a minimum of 3 times. Error measurements should be provided (not visible on all bars).*

Response: Data are for at least 3 independent biological repeats (therefore assays performed a minimum 3 times). In light of data transparency, individual data points were included on histograms around the mean (±S.E.) values to show variability.

7) *Line 178: since no target validation was performed, conclusions re: DHFR in gametocytes is premature based on the provided data alone.*

Response: This sentence has been modified to clearly indicate correctly per the reviewer's comment that our data simply supports previous reports where DHFR is not active in gametocytes and nor are any other antifolates active.

REVIEWER 2

The need for novel scaffolds targeting multiple plasmodial life cycle stages is a high priority in the backdrop of resistance developing against front line antimalarials.

The manuscript presents a well-designed parallel screening of the MMV-PRB compounds for potential antimalarial activity against asexual PfNF54, stage IV/V gametocytes from PfNF54 and P. berghei liver stages.

The data is presented in a logical and coherent manner and the study design has built in successful validations to corroborate the results presented.

Questions:

-While the supplementary data does provide detailed info for the full screen, it would be good to add in a 'filtered file' containing the selection criteria and relevant data for the post-cytotoxicity filtered panel. It is difficult to ascertain on which compounds the CC50 (HepG2) and viability (CHO) assays were done and the corresponding selectivity indices. Perhaps a more focussed presentation of this data in the manuscript will enable the reader to grasp the major finding of the study more readily.

Response: We appreciate this comment and concur that a more 'user-friendly' filtered file could be useful. We therefore included a filtered dataset as a second tab in the supplementary data file (supplementary file 2), that is

easily searchable through drop-down filters applied to all columns. For clarity, we included the selection and progression criteria as well. We also included selectivity indices as requested.

-Figure 6 is quite complex and difficult to follow. Could the authors consider an alternate mode of presentation of this data (preferably simplified).

Response: To address the reviewer's concerns, we simplified the histograms in Figure 6 and combined them with the new modified Figure 5. We retained only the carry-over EIA data and TRA SMFA data in the new histograms and provide the direct EIA and TBA SMFA as supplementary figures. In light of data transparency, individual data points were included on histograms around mean (\pm S.E.) values to show variability.

*-The detection of the two non-cytotoxic multistage PRB compounds (Birinapant and AZD-0156) and a range of dual active compounds is indeed exciting. However, the discussion then goes on to speculate potential hurdles for the progression of these leads (discrepancies with previous in vivo studies, in vitro cytotoxicity etc.). The value of this very well carried out study would probably be best realised if in-vivo screens and data on PK/PD and Tox are carried for the selected candidates with atleast dual stage effectivity. Having said the, **the vast amount of info contained in the manuscript will most certainly be beneficial if released into the wide research community and the manuscript merits publication, albeit in a revised format.***

Response: We simplified and clarified the discussion to not remove impact from the main contributions that the paper makes. We do agree that *in vivo* efficacy, PK/PD and tox data is important for selected candidates, and indeed the topic of current investigations. However, we concur with and appreciate that the reviewer indicates that the data provided in the paper is sufficient for publication and will benefit the research community.

REVIEWER 3

The present manuscript by Janette Reader et al. entitled "Multistage and transmission-blocking targeted antimalarials discovered from the open source MMV Pandemic Response Box" reports a considerable amount of work for the in vitro screening the Pandemic Response Box against several stages of the Malaria Parasite, P falciparum, with the intent to identify new lead compounds.

The work is well done, the methodology is appropriate. There are two positive aspects in the manuscript
1- parallel screening of the same compounds against different stages (asexual parasites, stage IV/V gametocytes, gametes, oocysts and liver stages) of *P. falciparum* and for endectocidal activity
2- the application of orthogonal assays against gametocytes which validate the results.

*Overall, the data are interesting since they contribute to explore different chemical spaces as sources of novel antimalarials, but also relatively disappointing since none of the active compounds shows an IC50 in the low nanomolar range, as expected for a promising lead, at least against the asexual stage. The only one was MMV1580844 (IC50 0.0017 μ M) a DHFR inhibitor, very active also against *P. berghei* liver stages, which unfortunately showed strong cross resistance with pyrimethamine. The most interesting compounds are those specifically and exclusively targeting stage IV-V gametocytes. Two of the most active, namely MMV1580843, MMV687273, have a known target (MmpL3) in bacteria, but their mode of action in *P. falciparum* has not been elucidated, yet. A deeper investigation of some of the most active compounds would made the manuscript more innovative.*

Response: We thank the reviewer for the points raised. We agree that the majority of the paper provided a descriptive dataset, but as Reviewer 2 indicated, we believe this to be novel enough to be of benefit to the wider research community, both against malaria as well as other infectious diseases. We believe this paper to have similar potential impact to previous papers published in Nature Comms (Miguel-Blanco et al. 2017, and Delves et al. 2018). Independent of the potent DHFR inhibitor, we do also provide novel chemotypes for asexual activity (13 compounds), which we believe will incite numerous downstream studies.

However, based on the reviewer's recommendation and to increase the innovation associated with the paper, we did follow the reviewer's suggestion to include mechanistic data on some of the most active compounds. We provide data for ML324 (the most potent gametocytocidal compound), showing this compound results in aberrant gene expression in gametocytes by changing histone methylation levels as inhibitor of a jmj demethylase in Plasmodia. Additionally, we provide the first investigations of the antitubercular clinical candidate MMV687273 (SQ109), showing that MmpL3 is not the target in Plasmodia, nor is other related RND superfamily members. We performed resistant mutant selection on SQ109-treated parasites (as is the norm in determining the molecular target of antimalarial drugs), which implicates a novel antimalarial target, the V-type H⁺-ATPase subunit A. We therefore believe that the paper now provides a powerful dataset to the community and additionally provides novel druggable processes worth further exploration. As indicated above, this information is now provided as new figures, Figure 6 and 7.

REVIEWERS' COMMENTS

Reviewer #1 (Remarks to the Author):

The investigators have addressed many of my prior concerns, and the additional methodological clarity and data transparency is appreciated. New mechanistic data is now provided for possible targets of ML324 (reported to inhibit Pfjnj3) and SQ109. The team identify transcriptional changes consistent with jmj3 inhibition in ML324-treated parasites. For SQ109, resistant parasites are identified with new mutations within the nucleotide binding pocket of VapA, a PM/DV H⁺-ATPase. A few additional concerns are noted related to the new studies included in this revision:

Major concerns:

1) The authors indicate that VapA is a "novel druggable target" of SQ109 (line 405). While caveat is included (line 409), unless additional data is provided to confirm cellular or in vitro VapA inhibition by SQ109, this conclusion should be further softened. There are many examples *P. falciparum* in which mutations in non-targets confer antimalarial resistance.

2) Supplemental Table S3: All SNPs in resistant strains should be provided, not just the SNPs in PF3D7_1311900

Trivial concern:

3) Lines 301, 394: Plasmodia is incorrect (PMID: 22738856). Consider as an alternative: Plasmodium spp.

Reviewer #2 (Remarks to the Author):

I am satisfied that the revisions requested have been addressed appropriately. I do believe that the current version of the manuscript is very much improved and suitable for publication.

Reviewer #3 (Remarks to the Author):

The manuscript by Janette Reader et al. entitled "Multistage and transmission-blocking targeted antimalarials discovered from the open source MMV Pandemic Response Box" has been corrected and resubmitted.

New data on the potential mechanism of action of two of the most active transmission blocking compounds found in the screening of the PRB have been provided. One is the antineoplastic epidrug ML324 (MMV1580488) which seems to inhibit the active site of Jumonji-domain containing demethylase (KDM4), Pfjnj3 thus preventing histone demethylation. The second hit is SQ109, which is known to inhibit MmpL3 in Mycobacteria, with no direct homology with Pf genes. However, by resistant mutant selection, the authors identified mutations in the *P. falciparum* V-type H⁺-ATPase, an enzyme shown to be present in the parasite plasma membrane and in the digestive vacuole, where it regulates H⁺ efflux and pH. The data are convincing and open the way to further investigation on PfVapA as a novel druggable target for late stage gametocytes. In this format, the paper seems significantly improved and suitable for publication.

Minor comments

Line 204-206 - the compounds MMV1580496 (triapine) and the bacterial methionyl-tRNA synthetase

inhibitor MMV1578884 (REP3123) are not indicated in fig4

Line 216. There is a minor typo with fig.5 MMV687273, IC50 = 0.105 μ M. The same compound in fig 5 shows IC50 = 0.107 μ M

Line 223-226 Among the 8 compounds that inhibited male exflagellation by $\geq 80\%$, one is the MmpL3 inhibitor MMV1580843. Please, note that in figure 5, panel B, there is probably a typo between compound MMV1580843 and compound MMV1580483 (AZD) which shows only 60% inhibition of male exflagellation.

Multistage-active and transmission-blocking targeted antimalarials discovered from the open-source MMV Pandemic Response Box

Janette Reader^{1a}, Mariëtte E. van der Watt^{1a}, Dale Taylor², Claire Le Manach², Nimisha Mittal³, Sabine Otilie³, Anjo Theron⁴, Phanankosi Moyo¹, Erica Erlank⁵, Luisa Nardini⁵, Nelius Venter⁵, Sonja Lauterbach⁶, Belinda Bezuidenhout⁶, Andre Horatscheck², Ashleigh van Heerden¹, Natalie J. Spillman⁷, Anne N. Cowell³, Jessica Connacher¹, Daniel Opperman¹, Lindsey M. Orchard⁸, Manuel Llinás^{8,9}, Eva S. Istvan⁷, Daniel E. Goldberg⁷, Grant A. Boyle², David Calvo¹⁰, Dalu Mancama⁴, Theresa L. Coetzer⁶, Elizabeth A. Winzeler³, James Duffy¹¹, Lizette L. Koekemoer⁵, Gregory Basarab², Kelly Chibale^{2,12}, Lyn-Marié Birkholtz¹

Point-by-point responses to reviewers comments:

REVIEWER 1

The investigators have addressed many of my prior concerns, and the additional methodological clarity and data transparency is appreciated. New mechanistic data is now provided for possible targets of ML324 (reported to inhibit Pfjnj3) and SQ109. The team identify transcriptional changes consistent with jmj3 inhibition in ML324-treated parasites. For SQ109, resistant parasites are identified with new mutations within the nucleotide binding pocket of VapA, a PM/DV H⁺-ATPase. A few additional concerns are noted related to the new studies included in this revision:

Major concerns:

1) The authors indicate that VapA is a “novel druggable target” of SQ109 (line 405). While caveat is included (line 409), unless additional data is provided to confirm cellular or in vitro VapA inhibition by SQ109, this conclusion should be further softened. There are many examples P. falciparum in which mutations in non-targets confer antimalarial resistance.

Response: We agree with the reviewer’s comments and have softened the statement as requested to clarify that the mutations may simply be a resistance profile and further validation is required to confirm VapA as target and not just resistance mechanism.

2) Supplemental Table S3: All SNPs in resistant strains should be provided, not just the SNPs in PF3D7_1311900

Response: The information was included as requested.

Trivial concern:

3) Lines 301, 394: Plasmodia is incorrect (PMID: 22738856). Consider as an alternative: Plasmodium spp.

Response: Modified as requested

REVIEWER 2

I am satisfied that the revisions requested have been addressed appropriately. I do believe that the current version of the manuscript is very much improved and suitable for publication.

Response: We thank this reviewer and are happy that he/she is satisfied.

REVIEWER 3

The manuscript by Janette Reader et al. entitled “Multistage and transmission-blocking targeted antimalarials discovered from the open source MMV Pandemic Response Box” has been corrected and resubmitted. New data on the potential mechanism of action of two of the most active transmission blocking compounds found in the screening of the PRB have been provided. One is the antineoplastic epidrug ML324 (MMV1580488) which seems to inhibit the active site of Jumonji-domain containing demethylase (KDM4), Pfjnj3 thus preventing histone demethylation. The second hit is SQ109, which is known to inhibit MmpL3 in Mycobacteria, with no direct homology with Pf genes. However, by resistant mutant selection, the authors identified mutations in the P. falciparum V-type H⁺-ATPase, an enzyme shown to be present in the parasite plasma membrane and in the digestive vacuole, where it regulates H⁺ efflux and pH. The data are convincing and open the way to further investigation on PfVapA as a novel druggable target for late stage gametocytes. In this format, the paper seems significantly improved and suitable for publication.

Minor comments

Line 204-206 - the compounds MMV1580496 (triapine) and the bacterial methionyl-tRNA synthetase inhibitor MMV1578884 (REP3123) are not indicated in fig4

Response: We included these compounds in the figure as requested.

Line 216. There is a minor typo with fig.5 MMV687273, IC50 = 0.105 μ M. The same compound in fig 5 shows IC50 = 0.107 μ M

Response: This has been corrected.

Line 223-226 Among the 8 compounds that inhibited male exflagellation by $\geq 80\%$, one is the MmpL3 inhibitor MMV1580843. Please, note that in figure 5, panel B, there is probably a typo between compound MMV1580843 and compound MMV1580483 (AZD) which shows only 60% inhibition of male exflagellation.

Response: Thank you for picking up this typo, which is in the text itself, MMV1580843 has ~60% inhibition and MMV1580483 has ~90% inhibition, this has been corrected. The figure is correct.